# A hierarchical transcription factor cascade regulates enteroendocrine cell diversity and plasticity in *Drosophila*

Xingting Guo[1,2], Yongchao Zhang[1,2], Huanwei Huang[1,2] & Rongwen Xi [1,2] ✉

Enteroendocrine cells (EEs) represent a heterogeneous cell population in intestine and exert endocrine functions by secreting a diverse array of neuropeptides. Although many transcription factors (TFs) required for specification of EEs have been identified in both mammals and *Drosophila*, it is not understood how these TFs work together to generate this considerable subtype diversity. Here we show that EE diversity in adult *Drosophila* is generated via an "additive hierarchical TF cascade". Specifically, a combination of a master TF, a secondary-level TF and a tertiary-level TF constitute a "TF code" for generating EE diversity. We also discover a high degree of post-specification plasticity of EEs, as changes in the code—including as few as one distinct TF—allow efficient switching of subtype identities. Our study thus reveals a hierarchically-organized TF code that underlies EE diversity and plasticity in *Drosophila*, which can guide investigations of EEs in mammals and inform their application in medicine.

Enteroendocrine cells (EEs) are a heterogeneous population of cells generated from intestinal stem cells that are scattered throughout the intestinal epithelium, with subtypes distributed in a spatially-restricted pattern along the length of gut and along the crypt-villus axis. They are considered as the major sensor of luminal content, and are known to respond to stimuli by secreting various neuropeptides/hormones to regulate diverse physiological processes, such as appetite, gut motility, glucose and lipid homeostasis[1,2]. The clinical relevance of specific EE-secreted hormones and EE subtypes has been highlighted recently: mimetics of the L-cell-produced Glucagon-like Peptide-1 (GLP-1) have been applied for the clinical treatment of diabetes and obesity[3,4]. However, the process during which EE cell diversity is generated is still not well-understood.

EEs in the adult *Drosophila* midgut show great similarity with mammals in their function, spatial distribution, cellular diversity, and cell lineage origins[5]. *Drosophila* EEs are distributed in the intestinal epithelium in a scattered pattern along the length of the midgut, constituting ~10% of cells in the epithelium[6,7]. These EEs have long been considered as heterogeneous, based on the observation that many EE-secreting hormones are expressed only in a subset of EEs and/ or at

specific gut regions[8,9]. The ongoing shift toward single cell level analysis has enabled the characterization of EE diversity in exquisite detail: to data, *Drosophila* EEs are known to compromise 10 major subtypes that are all regionally-distributed and can be largely classified into two major groups: class I subtype group that includes I-a, I-ap-a/p, I-m, I-p^CCHa1 and I-p^AstA subtypes and class II EE subtype group that includes II-a, II-m1, II-m2 and II-p subtypes, in addition to a minor class III subtype[10]. The class I and II subtypes are distinguished by their mutually exclusive expression of neuropeptides Allatostatin C (AstC) and Tachykinin (Tk), respectively, and each of the 10 subtypes expresses ~1 to 4 additional neuropeptide genes[10]. For simplicity, the subtypes within the class I or II groups are hereafter referred to as subclass subtypes.

Similar to mammals, *Drosophila* EEs are periodically generated from intestinal stem cells[6,7]. The specification of EEs from intestinal stem cells is initiated by a transient expression of TF Scute, which induces asymmetric stem cell division to generate EE progenitors (EEPs), each EEP then divides typically once before the accumulation of the EE fate determination factor Prospero (Pros) and consequent execution of terminal differentiation[11]. The division of EEPs is

[1]National Institute of Biological Sciences, No. 7 Science Park Road, Zhongguancun Life Science Park, 102206 Beijing, China. [2]Tsinghua Institute of Multi-disciplinary Biomedical Research, Tsinghua University, 102206 Beijing, China. ✉e-mail: xirongwen@nibs.ac.cn

asymmetric and the resultant two EE daughters respectively express AstC and Tk, which mark class I and class II EEs, respectively[11,12]. The asymmetric division of EEP appears to be regulated by differential Notch activity, as loss of Notch causes failed specification of Tk[+] class II EEs[12]. The single cell transcriptome analysis of EEs has also facilitated the identification of an array of TFs that are expressed in each EE subtype, and subsequent genetic analyses have delineated which TFs are required for the specification of each known EE subtype. For example, Ptx1 and Mirr are respectively required for AstC[+] class I and Tk[+] class II EE subtypes, and additional region specific TFs, including the middle midgut enriched Esg and posterior midgut enriched TF Drm are known to further contribute to the specification of sub-class EE subtypes[10].

In this work, we investigated the regulatory and functional relationships among these EE-specific and EE-subtype-specific TFs, and how these TFs work together to generate EE cell diversity, we first investigated the identity maintenance function of Pros for EEs, and the genome-wide analysis of Pros target genes. Followed by enhancer activity analysis and functional genetics, our results collectively suggest a master factor (Pros) based- "TF cascade" in which a top-down hierarchy of TFs forms an additive TF cascade to generate EE subtype diversity. We also observe a surprising degree of cellular plasticity of the differentiated EEs, reflected by efficient identity switch among distinct EE subtypes via conditional gain or loss of a single TF.

## Results

### Depleting *pros* in mature EEs leads to loss of EE identity

Pros, a homeodomain related TF previously implicated in neuronal cell differentiation[13], was initially used as a pan-EE cell marker in the midgut[6,7]. Later it was demonstrated to be essential for the differentiation of EEs: its loss causes a failure of EE generation from intestinal stem cells, whereas its ectopic expression in intestinal stem cells is sufficient to induce stem cell differentiation into EEs[14–16]. Among TFs that have been implicated in the regulation of EE specification from intestinal stem cells, including Esg, the achaete-scute gene complex (AS-C) and Ttk, Pros acts at downstream of all these factors[14,17–19]. Therefore, Pros appears to be the key regulator downstream of multiple signaling pathways to promote the specification of EEs from intestinal stem cells. However, despite its importance in EE specification, it is unclear whether Pros is continuously required in differentiated EEs for maintaining EE identity.

To address this question, we specifically depleted *pros* in EEs in prosV1-Gal4, gal80[ts]; UAS-GFP adult flies for 7 days and examined the expression of AstC and Tk, which mark class I and class II subtypes, respectively. We found that despite the total number of GFP[+] cells did not alter significantly, virtually Tk and AstC expression were all completely lost in these *pros*-depleted GFP[+] cells (Fig. 1a–i). The retaining of GFP expression indicates that the conditional loss of *pros* does not affect cell survival, neither the transcriptional activity of the *pros* promoter, whereas the loss of hormone expression indicates that the conditional loss of *pros* is sufficient to compromise hormone production in both class I and class II EE subtypes. As Pros expression is initiated at early EEP stages, to exclude the possibility that the loss of hormone expression is due to blocked EE differentiation from progenitor cells, we performed a time lapse analysis, and observed a gradual loss of Pros expression in the differentiated EEs that is companied by a gradual loss of hormone expression over 1, 3, and 7 days following *pros* depletion. Similarly, the number of GFP[+] cells remains constant during this process (Supplementary Fig. 1a–d, i, j), suggesting that the cells are not disappeared. Staining with an apoptotic marker Dcp-1 also revealed that there was no obvious increase in the incidence of cell death during the process (Supplementary Fig. 1g, h). Therefore, the depletion of pros in the differentiated EEs does not affect cell survival, at least during this experimental time window, but rapidly compromises EE identity, as reflected by failed hormone production.

Notably, when the flies treated for 7 days were shifted back to permissive temperature, the expression of Pros and hormones were quickly restored, showing that the loss of EE cell identity by Pros depletion is reversible (Supplementary Fig. 1e, f). These observations collectively demonstrate that Pros is continuously required for maintaining the identity but not the survival of the differentiated EEs.

### Depleting *pros* in mature EEs lead to global loss of EE-specific transcription programs

To further investigate the global effect of *pros* depletion on EE cell identity, we determined the gene expression changes. To do this, 5–7 day old adult female flies of prosV1-Gal4[ts],UAS-GFP and prosV1-Gal4[ts],UAS-GFP,UAS-*pros-RNAi* that were cultivated at permissive temperature were collected and shifted to restrictive temperature for 7 days. Subsequently, we FACS-sorted both control and *pros*-depleted EEs and performed bulk RNA-sequencing analysis (3 independent replicates for both control and experimental groups). As is shown in the volcano plot (Fig. 1j), *pros*-depletion led to significant upregulation of 1741 genes and downregulation of 1051 genes when compared to control EEs (Supplementary Data 1). By comparing to our previously-reported transcriptome of esg+ progenitor cells and enterocytes in the epithelium[20,21], which were acquired based on a common experimental procedure, we selected top 250 EE-enriched genes (noted as EE signature genes, Supplementary Data 2) and performed gene set enrichment analysis (GSEA) of the genes that were significantly altered by *pros* depletion. Almost all EE signature genes were significantly down-regulated following *pros* depletion, with a normalized enrichment score (NES) of −3.466 (Fig. 1k). Gene ontology analysis also revealed that all of the top 10 gene categories of down-regulated genes were closely related to various aspects of cellular functions specific to EE, such as neuropeptide-GPCR signal transduction, ion channels, transmembrane transport, and neurotransmitter secretion (Fig. 1l and Supplementary Data 3). In addition, many EE signature genes, such as hormones and GPCRs, including hormones that distinguish EE subtypes (such as DH31 for IIp subtype and sNPF for class III subtype), and genes involved in post-transcriptional regulation and secretion of peptide hormones were significantly down-regulated upon *pros* depletion (Supplementary Fig. 2a–d). By immunostaining, we further confirmed the reduced expression of a pan-EE marker Rab3 (Supplementary Fig. 2e, f), as well as a hormone receptor CCHa1-R, which is normally expressed in EEs at anterior and middle midgut regions, following pros depletion (Supplementary Fig. 1g, h). However, we did observe a few exceptions: peptide hormones ITP and Nplp2, as well as a GPCR receptor Tre1 instead showed increased expression following *pros* depletion (Supplementary Fig. 2a, c, highlighted in red). Interestingly, the expression of these genes is not restricted to EEs[10,22]. More specifically, Tre1 is highly expressed in progenitor cells, and ITP is expressed in all intestinal epithelial cells at the posterior end of the fly midgut[10]. Therefore, Pros regulates the expression of the entire peptide hormones and GPCRs that are specifically expressed in EEs or EE subtypes in the intestinal epithelium.

EE diversity is dependent on subtype-specific expression of many TFs[10]. To investigate whether these TFs are also regulated by Pros, we profiled their expression in normal and *pros*-depleted EEs. The results showed that most of these TFs were significantly downregulated upon *pros-IR*, including Ptx1 and Mirr, which are respectively required for specification of class I and class II subtypes, as well as subclass-specific TFs, including Drm, Hbn, Poxn, Fer1, Sug and Dac. Surprisingly, a number of TFs that are known to be required for the specification of EE subtypes, including Esg, Nlp, Mamo, Exex and NK7.1 (highlighted by green), were up-regulated following *pros* depletion (Supplementary Fig. 2J). Interestingly, the expression of these TFs again is not restricted to EEs, as they are also expressed in intestinal progenitor cells and/or enterocytes (Supplementary Fig. 2I). Therefore, Pros regulates the

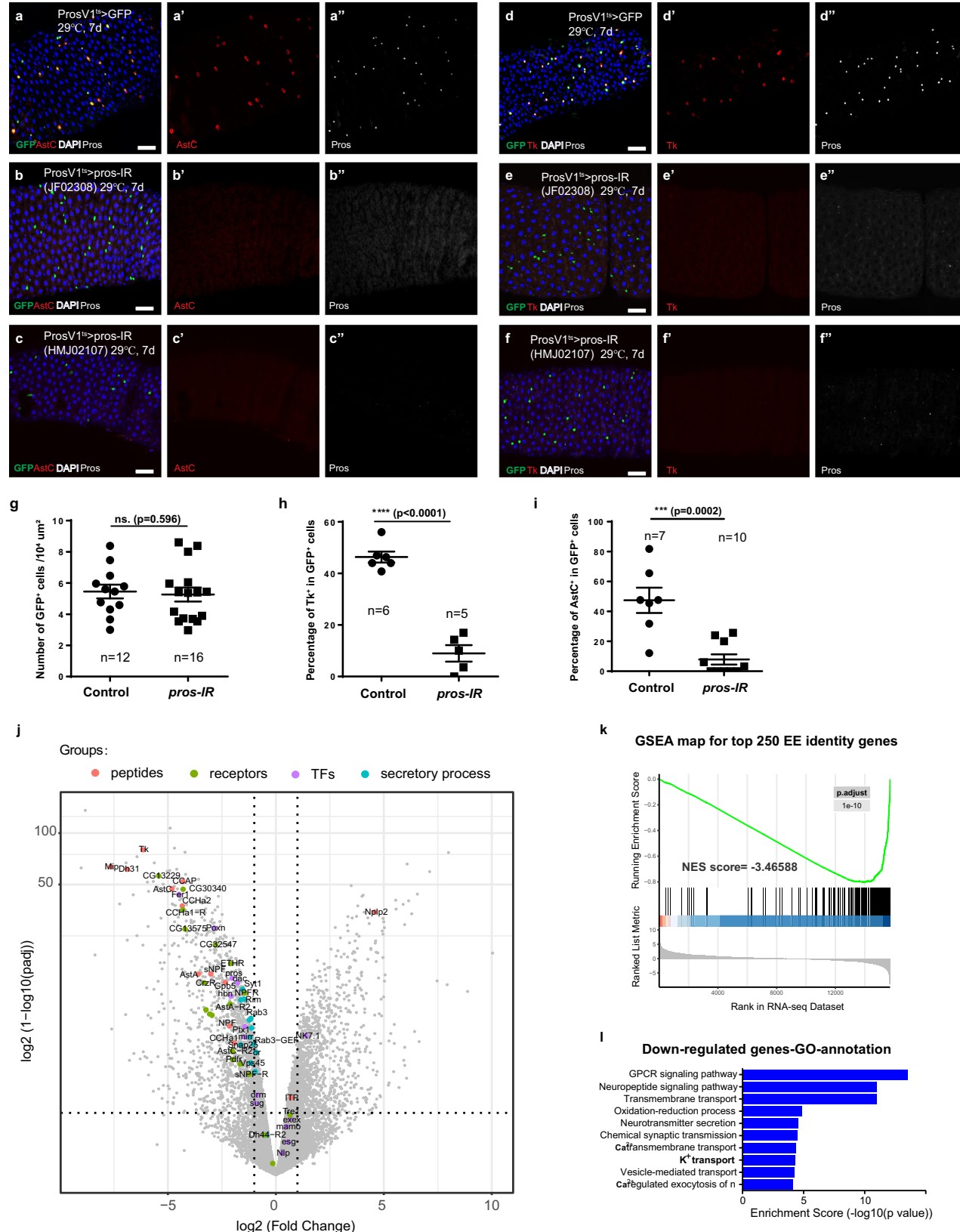

expression of all EE-specifically-expressed TFs and peptide hormones, further supporting the idea that Pros is a master regulator of EE fate.

## Pros directly targets EE identity genes revealed by DamID

It is known that Pros regulates the neuroblast-to-neuron differentiation process by directly repressing cell cycle and neuroblast-specific genes and activating neuron-related genes to promote neuron differentiation[23], we thus suspect that Pros may directly target and promote transcription of EE-specific genes to promote EE specification. We therefore performed a targeted DNA Adenine Methyltransferase Identification (DamID) by specifically expressing optimized Dam (oDam) alone (control group) or oDam-Pros fusion

**Fig. 1 | Pros maintains mature EE identity. a–f** Neuropeptides AstC (**a**) and Tk (**d**) marks two major classes of EEs, and knocking down *pros* in EEs by shifting adult F1 progeny to 29 °C for 7 days abolishes both AstC (**b, c**) and Tk (**d, e**) expression. Two independent pros-RNAi lines were used: **b–e** for JF02308; **c–f** for HMJ02107; Quantification of total GFP⁺ cell number per $1 \times 10^4 \mu m^2$ (**g**), percentages of Tk⁺ cells (**h**) and AstC⁺ cells (**i**) in GFP⁺ cells of control and *pros*-depleted guts, Error bars represent Mean ± SEM, ns not significant, ***$p < 0.001$, ****$p < 0.0001$ (two-tailed Student's *t* test); "n" indicate the number of guts used for quantification. Source data are provided as a Source Data file. **j** Volcano plot showing the transcriptome comparison between control and *pros*-depleted EEs. Most members within various EE signature gene sets, including neuropeptides (orange spots), peptide receptors (green spots), neuropeptide regulating TFs (purple spots) as well as factors involved in the secretory process (cyan spots), are significantly down-regulated upon *pros*-depletion. Statistical test of *p* value: two-tailed Wald test with adjustments. **k** GSEA plot showing transcriptional alternations of EE signature genes. Signature genes for certain cell type were selected according to both absolute gene expression value and comparative enrichment compared to other cell types. Among the top 250 EE signature genes, most of them are significantly down-regulated upon *pros*-depletion, with an NES score of −3.46588; Statistical test of *p* value: empirical phenotype-based permutation test. **l** Gene ontology (GO) of significantly downregulated genes upon *pros* depletion. Top 10 most enriched GO terms of down-regulated genes are displayed; Statistical test of *p* value: modified Fisher's exact test (EASE score). Scale bars, 50 μm.

transgene (experimental group, Fig. 2a) in EEs to profile potential Pros targets according to the previously described iDamID/iDEAR pipeline[24]. Ectopic expression of Dam-Pros in progenitor cells led to near complete depletion of these cells, similar to that caused by overexpressing Pros alone[14], indicating that the chimeric protein is biologically functional (Supplementary Fig. 3a, b). Two replicates for Dam-ID experimental group and three for control groups were carried out, and Pearson co-relation scores revealed excellent consistency inside groups and significant differences between control and experimental groups (Supplementary Fig. 3c). Parameters for calling Pros binding peaks as positive peaks were set as the following: the log 2FC between Dam-pros and control is above 1 and the adjusted *p* value is below 0.01. In total, we observed 2936 significantly enriched Pros binding peaks, corresponding to 2238 different genes.

We next performed a combined analysis of the Dam-ID and RNA-seq results to identify genes that are both DamID- targeted and transcriptionally altered upon *pros* depletion, and these genes were then considered as Pros-target genes. With this approach, we identified 690 such Pros target genes, with 309 of them showed significant downregulation upon *pros* depletion, while the rest showed significant upregulation upon *pros* depletion (Supplementary Fig. 3d). The comparable number of up- and down-regulated genes suggests that Pros may function either as a suppressor or an activator of transcription, an idea that has been proposed previously[23]. Further GSEA analysis revealed that 68 of the top 250 EE signature genes were targeted by Pros, and almost all of these genes were down-regulated upon *pros*-depletion (Fig. 2b and Supplementary Data 4). GO category analysis of the Pros target genes highlights many EE-specific gene programs, including neuropeptide and GPCR signaling, transmembrane transport, as well as ion channel activity (Fig. 2c and Supplementary Data 5). By plotting these Pros target genes in a scatter graph, we readily observed many genes that are related to EE subtypes, such as neuropeptide AstA that is expressed exclusively in class I group subtypes, Tk and NPF that are exclusive to class II group subtypes, as well as a number of TFs that are involved in regulating EE subtype specification, such as Mirr, Ptx1, and Fer1 (Fig. 2d and Supplementary Data 6).

To further validate the Pros binding sites on the Pros target genes identified by DamID, we selected two neuropeptide genes and constructed LacZ reporter lines driven by the putative Pros binding regions, and named the reporter lines as CCHa1^Pros-lacZ and NPF^Pros-lacZ, respectively. The neuropeptide CCHa1 is normally expressed in several EE subtypes across class I and class II groups, and the putative binding site is localized within the first intron of the CCHa1 locus (Fig. 2e); The neuropeptide NPF is specifically expressed in several subtypes specifically within the class II group, and the putative Pros binding region is about 2 kb upstream of the transcription starting site (TSS) (Fig. 2h). Both CCHa1^Pros-lacZ and NPF^Pros-lacZ showed specific expression pattern in a subset of EEs (Fig. 2f, j, yellow arrow heads), and in both cases, the expression was diminished upon *pros* depletion (Fig. 2g, j, white arrow heads). A consensus Pros binding motif (T-A/T-A-G-A/C/G-C-G/A/T) has been previously defined[23,25], and interestingly,

we identified one putative Pros binding motif in the CCHa1 enhancer (+1734 bp to +1740 bp of the TSS), and two putative Pros binding motifs in the NPF enhancer (Motif 1: TTAGCCG, −1595 bp to −1589 bp; Motif 2: TAAGCTG, −1471 bp to −1465 bp of the TSS) (Fig. 2e, i). To determine if the putative Pros motif is important for the enhancer activity, we generated lacZ reporter lines with the same enhancer fragments except that the Pros binding motifs were deleted. As is shown in Fig. 2h, l, deletion of the Pros motif abolished the expression of both reporters in EEs. These results suggest that Pros directly regulates the enhancer activity of neuropeptide genes CCHa1 and NPF through the canonical Pros binding motif.

Mirr is a regulator for all class II subtypes, and a putative Pros binding region was found at about 1 kb upstream of the TSS site, close to the previously reported mirr^B1-B12-lacZ enhancer trap insertion site (Fig. 2m)[26]. This reporter was specifically expressed in Tk⁺ EEs, and similarly, its expression was readily abolished upon *pros* depletion (Fig. 2n, o). Ptx1 is a regulator for all class I subtypes, and a putative Pros binding region was found at the intronic region (Fig. 2p). The reporter driven by this region (Ptx1^Pros-lacZ) was found to be expressed in a subset of EEs, and its expression was also abolished upon *pros* depletion (Fig. 2q, r). Collectively, these results suggest that Pros directly regulates the transcription of EE-specifically-expressed TFs and neuropeptides, thereby further supporting its role as a master fate inducer and maintenance factor for the EE identity.

To facilitate the study of the regulatory relationships among subtype regulators, we hereafter consider Pros at the top of the TF hierarchy named as the 1⁰ TF, the TFs Mirr and Ptx1 that regulate the entire class I or class II subtypes are considered as class-level TFs and noted as 2⁰ TFs, and other TFs that regulate EE subtypes within either class I or class II groups are noted as 3⁰ TFs.

## Cell identity conversion between class I and class II subtypes upon gain or loss of Mirr

We next investigated the regulatory relationships between the 2⁰ TFs in determining class I versus class II subtype identity. Due to regional variations in the percentage of Tk⁺ or AstC⁺ EEs along the length of midgut, here we focused on R4c and R5 regions at the posterior midgut, where all EEs are either Tk or AstC positive with a 1:1 ratio (Fig. 3a, b). In agreement with a known essential role for *mirr* in class II EE specification, depletion of *mirr* in EEs with two independent RNAi lines all caused near complete loss of Tk expression in all EEs, while the expression of the 1⁰ TF Pros was not affected (Fig. 3c *mirr-RNAi* using JF02196 and Supplementary Fig. 4a *mirr-RNAi* using SH05171.N). Using both Gal4 and lexA expression systems, we further confirmed a requirement for *mirr* in the expression of DH31, another neuropeptide that is only expressed in several subtypes within the class II group[10] (Supplementary Fig. 5a, b). Surprisingly, following *mirr* depletion, we observed that almost all EEs turned on the expression of AstC, the marker for class I subtypes (Fig. 3b, d, i and Supplementary Fig. 4b). In addition, AstA, another neuropeptide that is only expressed in several subtypes within the class I group[10], was also induced in more than 80% EEs at the posterior midgut (Supplementary Fig. 5e, f, h). These

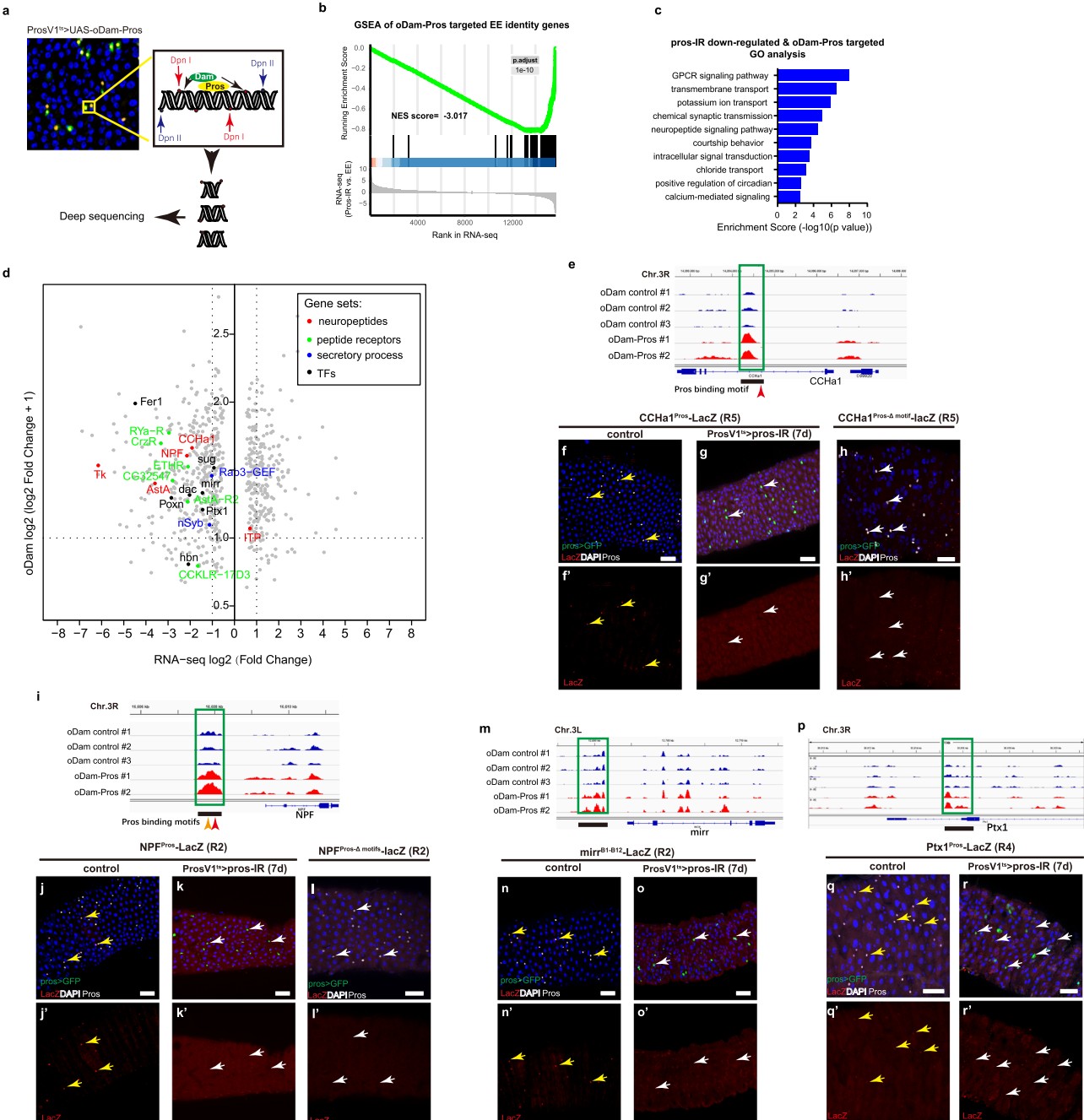

**Fig. 2 | Pros globally binds to and regulates the transcription of EE signature genes. a** A schematic diagram for Dam-ID analysis to seek for Pros binding sites in EEs of adult flies; **b** GSEA map of Dam-Pros targets among top 250 EE identity genes. Almost all of these genes are down-regulated upon *pros* depletion; **c** GO analysis of genes both targeted by Pros Dam-ID and down-regulated upon *pros* depletion; **d** Scatter graph showing multiple EE identity gene sets that downregulated after *pros-RNAi* were also targeted by Pros, including neuropeptides (orange spots), peptide receptors (green spots), and neuropeptide regulating TFs (purple spots). **e** Examples showing direct Pros targeting on neuropeptide gene CCHa1, the putative Pros targeting region and predicted Pros binding motif are labeled; A LacZ reporter driven by the putative Pros binding sequence on CCHa1 locus marks a subset of R5 EEs (**f**, yellow arrow heads), and depleting *pros* by ProsV1ts > *pros*-RNAi abolishes LacZ expression entirely (**g**, white arrow heads), depleting the conserved Pros binding motif significantly abolished lacZ activity (**h**, white arrow heads);

**i** Examples showing direct Pros targeting on neuropeptide gene NPF, and the putative Pros targeting region and predicted Pros binding motif are labeled; A LacZ reporter driven by the putative Pros binding sequence on NPF locus marks a subset of R2 EEs (**j**, yellow arrow heads), and knocking down *pros* abolishes LacZ expression entirely (**k**, white arrow heads), depleting the two conserved Pros binding motif also significantly abolished lacZ activity (**l**, white arrow heads); **m** An example showing direct Pros targeting on gene locus of mirr. The putative Pros targeting region is underlined. Mirr-lacZ is specifically expressed in a subset of EEs (**n**), and its expression is abolished by knocking down *pros* (**o**). **p** Examples showing direct Pros targeting on gene loci of Ptx1. The putative Pros targeting region is underlined; A LacZ reporter driven by the putative Pros binding sequence on Ptx1 locus marks a subset of EEs located at R4 region (**q**, yellow arrow heads), and *pros* depletion entirely abolishes LacZ expression (**r**, white arrow heads), Scale bars, 50 μm.

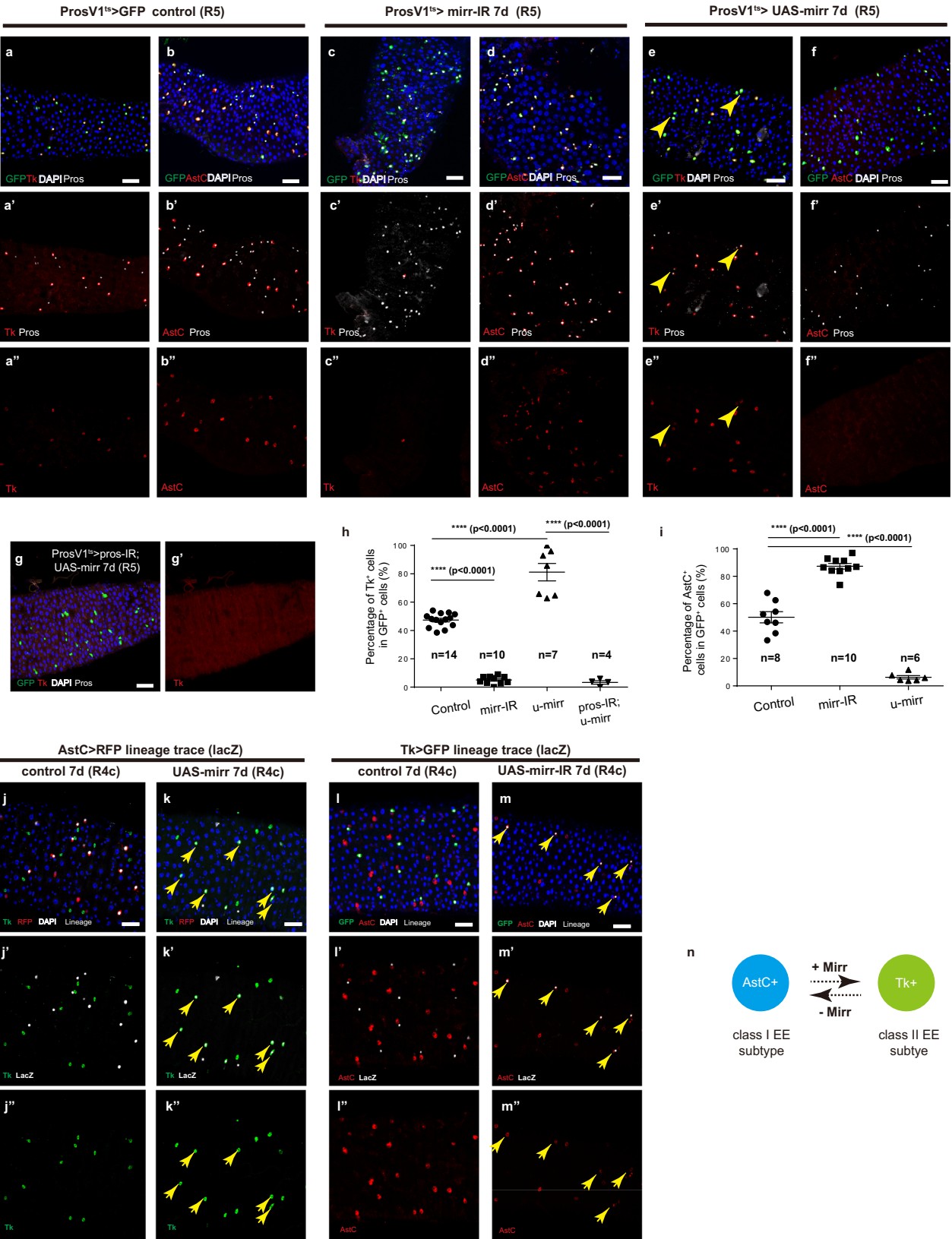

observations indicate a possibility that depletion of *mirr* induces a fate switch from class II to class I subtype identity.

To further test the idea of cell fate conversion, we overexpressed *mirr* in all EEs for 7 days, and observed that class II subtype-specific neuropeptides Tk and DH31 were expressed in virtually all EEs at the posterior midgut (R5 region) (Fig. 3e, h and Supplementary Fig. 5c, d),

while the expression of class I- specific neuropeptides AstC and AstA were abolished (Fig. 3f, i and Supplementary Fig. 5g, h). These phenomena cannot be simply explained by the loss of pre-existing EEs and regeneration of new EEs from intestinal stem cells. Normally the stem cells are relatively quiescent and the epithelium turns over in 2 or more weeks[27,28]. In addition, altering *mirr* overexpression does not appear to

**Fig. 3 | Subtype identity conversion between class I and class II EEs by gain or loss of Mirr. a–f** Tk marks class II subtypes (**a**), and AstC marks class I subtypes (**b**). Within R4c and R5 regions of the midgut, knocking down *mirr* by ProsV1ts > *mirr-RNAi* (JF02196, Chr.3) abolishes Tk expression (**c, h**), while turns on AstC expression in almost all EEs (**d, i**). Mirr overexpression by ProsV1ts > UAS-*mirr* activates Tk expression in almost all EEs (**e**, yellow arrowheads labels Tk+ EE pairs), while suppresses AstC expression (**f, i**); **g** despite ectopic expression of Mirr, Tk expression is not detected in the absence of *pros*; **h** percentages of Tk+ cells among all GFP+ cells in *control*, *mirr-IR*, *u-mirr* and *u-mirr; pro-IR* guts. Error bars represent Mean ± SEM, ****$p < 0.0001$ (two-tailed Student's *t* test), "n" indicate the number of guts used for quantification; Source data are provided as a Source Data file. **i** Percentages of AstC+ (I) cells among all GFP+ cells in control, *mirr-IR* and *u-mirr* guts. Error bars represent Mean ± SEM, ****$p < 0.0001$ (two-tailed Student's *t* test), "n" indicate the number of guts used for quantification; Source data are provided as a Source Data file. **j** AstC-gal4,u-RFP/cyo; Act«lacz,gal80ts and UAS-flp,Gal80/CyO are used to perform lineage tracing of AstC+ EEs. In normal guts, AstC-Gal4 lineage cells remain positive for AstC and negative for Tk. **k** Lineage tracing of AstC+ EEs in *mirr* overexpressed guts. AstC-Gal4 lineage positive cells have turned on Tk (yellow arrow heads), while the expression of AstC is inhibited. **l** UAS-flp,Gal80/CyO; dtk-gal4,UAS-GFP/TM6 and Act«lacz,gal80ts are used to perform lineage tracing of Tk+ EEs. In normal guts, Tk-Gal4 lineage cells remain positive for Tk and negative for AstC. **m** Lineage tracing of Tk+ EEs in *mirr*-depleted guts. Tk-Gal4 lineage cells have turned on the expression of AstC (yellow arrow heads), while the expression of Tk was inhibited. **n** A schematic model of identity conversion between class I and class II subtypes. In differentiated EEs, ectopic activation or knocking down *mirr* causes subtype identity conversion from class I to class II or class II to class I subtypes, respectively. Scale bars, 50 μm.

affect EE survival, as there was no obvious increase in cell death incidence following either *mirr* overexpression or depletion (Supplementary Fig. 6a–c). These observations suggest that Mirr is both necessary and sufficient for the establishment of class II EE subtype identity, and indicate that class I and class II EEs are probably interswitchable.

The function of the other $2^0$ TF Ptx1 appeared to be very different. Despite the fact that Ptx1 is required for AstC expression in class I subtypes, *ptx1* depletion in EEs failed to induce Tk expression (Supplementary Fig. 7a, b). Additionally, overexpression of *ptx1* in all EEs did not cause any visible alterations in neuropeptide expression patterns, neither AstC was ectopically activated nor Tk was suppressed (Supplementary Fig. 7c, d). Collectively, these observations suggest that during EE cell specification, the Ptx1-mediated specification of the class I EE acts as a default type, while the expression of Mirr in half of the committed progenitor cells allows the adoption of class II EE cell identity.

Next, we performed cell lineage tracing analysis to further confirm the direct identity conversion between the two classes of EEs. We crossed AstC-gal4,UAS-RFP/cyo; Act < stop < lacZ, tub-Gal80ts flies with UAS-flp,tub-Gal80ts or UAS-flp,tub-Gal80ts; UAS-*mirr* flies to perform cell lineage tracing of AstC+ EEs, and crossed UAS-flp,tub-Gal80ts/CyO; dtk-Gal4,UAS-GFP/TM6 flies with Act < stop < lacZ, tub-Gal80ts or UAS-*mirr-RNAi*; Act < stop < lacZ, tub-Gal80ts flies for tracing Tk+ EEs. Adult F1 progenies with desired genotypes were shifted to the restrictive temperature for 7 days before analysis. In these control guts, all AstC-Gal4 lineage positive class I subtypes were still AstC+ (visualized by AstC-Gal4 > RFP expression), and its expression was mutually exclusive with Tk (visualized by anti-Tk staining) (Fig. 3j). Similarly, the class II subtypes labeled by Tk-Gal4 lineage showed persistent expression of Tk during the 7 day tracing period, and the Tk+ cells were mutually exclusive with AstC+ cells (Fig. 3l). These results suggest that there is no identity switch occurred between class I and class II EE subtypes under normal conditions. However, ectopically expressing *mirr* in the AstC-Gal4 lineage EEs suppressed AstC expression and turned on Tk expression (Fig. 3k, yellow arrowheads), while knocking down *mirr* in the Tk-Gal4 lineage EEs led to the loss of Tk expression and acquisition of AstC expression (Fig. 3m, yellow arrowheads). These results confirm a direct identity switch between the two mature EE subtype and Mirr as a master regulator of class II subtype identity: loss of *mirr* in class II subtype is sufficient to convert them into class I subtype, and ectopic expression of *mirr* in class I subtype is sufficient to convert them into class II subtype identity (Fig. 3n).

### Pros cooperates with $2^0$ TFs to define class I and II EE subtypes

Having established a general requirement for Pros in EE identity maintenance, and an instructive role for Mirr in specifying class II subtype identity, we next determined whether Pros remains to be indispensable for class I and II subtype specification once Ptx1 or Mirr

expression has been activated. To do this, we overexpressed *mirr* and simultaneously depleted *pros* in EEs, and found that in this case, Tk expression was not induced (Fig. 3g, h), and in fact, the expression of AstC was lost as well, even under conditions when *ptx1* was overexpressed (Supplementary Fig. 7e). Thus, Pros is continuously required for both Mirr-mediated establishment of class II subtype identity and Ptx1-mediated establishment of class I subtype identity. Considering both the $1^0$ TF Pros and the $2^0$ TFs Mirr and Ptx1 are required for the enhancer activity that drives expression of neuropeptide genes (shown later in this study), it is tempting to speculate that Pros acts cooperatively with the $2^0$ TFs Mirr or Ptx1 on gene enhancers to promote transcription of their shared target genes.

### Notch acts at the early progenitor stage to regulate *mirr* expression and class II EE specification

Given that the expression of Ptx1 and Mirr both require Pros, how is the expression diversity achieved among EEs? Notch signaling is known to be specifically required for class II subtype specification from intestinal stem cells[12]. However, either ectopic activation or inhibition of Notch activity in the differentiated EEs fails to induce the class I and class II identity switch[12]. To understand the relationship between Notch and *mirr* during the specification of class II EE subtypes, we first determined whether Notch is required for *mirr* expression during EE specification. Knocking down *Notch* in progenitor cells using esg-Gal4ts > *notch-RNAi* for 3 days led to generation of EE cell clusters, and all these EEs were AstC+ class I subtypes and negative for Tk expression (Fig. 4a–d, yellow arrow heads). Mirr expression was also absent in the EE cell clusters generated by *Notch*-RNAi, as indicated by the absence of mirr-lacZ expression in Flp-out clones of *Notch-RNAi* (Fig. 4e, white arrowhead). Upon canonical Notch signaling activation, Suppressor of Hairless (Su(H)) functions as the core TF at the chromatin level to regulate transcription of Notch target genes. We therefore profiled a previously published Dam-Su(H) dataset and examined whether there is Su(H) binding activity in the regulatory region of *mirr*[20]. As is shown in Fig. 4f, a binding region of Su(H) was found at about 2 kb upstream to the TSS of *mirr* (highlight in red box), and this region also partially overlaps with the putative enhancer fragment targeted by Pros. These results collectively suggest that Notch signaling may directly regulate *mirr* expression to promote class II subtype specification (Fig. 4p).

Interestingly, we found that manipulating Notch activity in the differentiated EEs failed to alter the expression status of *mirr*. We overexpressed Nintra in AstC+ EEs using AstC-Gal4ts > UAS- Nintra for 7 days and this failed to induce mirr-lacZ expression; we depleted *Notch* in Tk+ EEs and this also failed to reduce mirr-lacZ expression in these cells (Fig. 4g, h, yellow arrowheads). Along with the previous observation that manipulating Notch activity in the differentiated EEs fails to alter neuropeptide expression patterns[12], these data collectively indicate that Notch activity is only transiently required at early progenitor stage for the induction of *mirr* expression and consequently

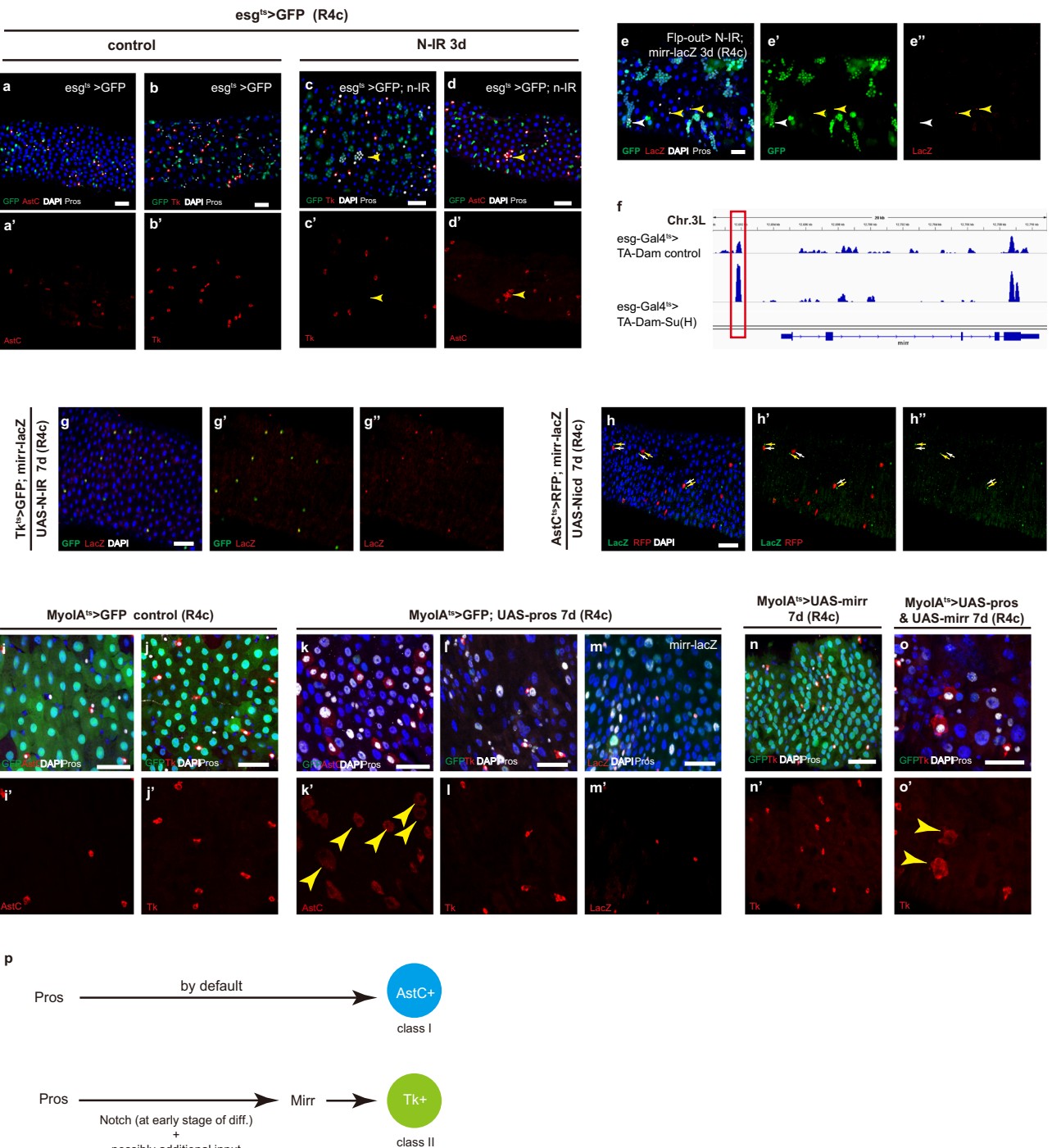

**Fig. 4 | The developmental regulation of class I versus class II subtype speci-fication. a–d** Knocking down *Notch* in progenitor cells using esg-Gal4ts leads to generation of Pros+ EE clusters expressing AstC (yellow arrowhead in **c**), but not Tk (yellow arrowhead in **d**); **e** the EE clusters in *Notch* mutant clones do not express *mirr* (white arrow head); **f** a graph showing enriched binding of Su(H) proximal to TSS site of *mirr* in progenitor cells, as highlighted in the red box. **g** Knocking down *Notch* in Tk+ class II subtypes fails to down-regulate mirr-lacZ expression (yellow arrow heads); **h** Ectopic activation of Notch signaling in AstC+ EEs fails to induce *mirr* expression (yellow arrow heads); In normal guts, AstC (**i**) and Tk (**j**) expression are restricted to Pros+ EEs; **k** ectopic expression of Pros in ECs activates AstC expression (yellow arrow heads); **l** ectopic expression of Pros does not activate Tk expression ECs; **m** ectopic expression of Pros in EC cells fails to induce *mirr* expression; ectopic expression of Mirr in ECs fails to activate Tk expression in ECs (**n**), while simultaneously overexpression of both Pros and Mirr induces Tk expression in ECs (**o**, yellow arrow heads); **p** a schematic model of cell fate deter-mination between class I and class II EE subtypes. During EE fate specification from EE progenitor cells, the class I fate is served as a default mode promoted by Pros, while acquisition of class II fate requires additional Notch activation, which must be available at early progenitor stage to allow Mirr expression and consequently the adoption of class II subtype identity. Scale bars, 50 μm.

class II subtype specification (Fig. 4p). Alternatively, there might be additional mechanisms involved for the initiation of *mirr* expression during class II EE specification, in addition to Notch.

## Ectopic *pros* overexpression in enterocytes induces AstC expression

Having established Pros as a master inducer of EE fate, we wondered whether ectopic expression of *pros* is able to induce neuropeptides, and if so, what kinds of neuropeptides can be induced. We ectopically expressed Pros in the polyploid enterocytes using Myo1A-Gal4$^{ts}$, UAS-GFP for 7 days, and this caused a significant decrease of GFP level in enterocytes (Fig. 4k, l). As MyoIA is an enterocyte-specific gene, its downregulation indicates a suppressive role of Pros on enterocyte identity. Interestingly, staining with anti-Pros antibody revealed that the level of Pros accumulation among enterocytes was highly variable, with undetectable in some enterocytes, and highly accumulated in some others (Fig. 4k, l). This raises a possibility that *pros* could be subjected to post-transcriptional levels of regulation in these cells. Importantly, in the polyploid cells with high accumulation of Pros, the expression of AstC was frequently turned on (Fig. 4k, yellow arrowheads). However, the expression of Tk was not observed in these cells or any polyploid cells (Fig. 4l). Therefore, ectopic Pros expression is able to induce the activation of class I, but not class II EE subtype marker gene.

As implied earlier, the specification of class II subtype requires Mirr. We found indeed that Pros overexpression failed to induce mirr-lacZ expression in enterocytes (Fig. 4m). Consistent with a requirement for Pros in mirr-mediated class II EE specification, overexpression of mirr alone in enterocytes failed to turn on Tk expression (Fig. 4n). However, when Pros and Mirr were co-overexpressed in enterocytes, Tk expression was frequently turned on in the polyploid cells that had significant Pros accumulation (Fig. 4o). Along with the previously-established role for Notch, these observations collectively support the idea that during EE specification from intestinal stem cells, Pros induces class I subtype specification by default, while Notch activity that occurred at early EEP stage bifurcate the progenitor cell pool and allows half of the progenitor cells turn on *mirr* expression and consequently the adoption of class II subtype specification (Fig. 4p).

## Pros cooperates with 2⁰ and 3⁰ TFs to define subclass EE subtypes

Next, we investigated the regulatory and functional relationships among 1⁰, 2⁰, and 3⁰ TFs in determining subclass EE subtype identity. As described previously, Pros directly regulates the transcription of the two 2⁰ TFs mirr and Ptx1. Similarly, Pros may also directly regulate 3⁰ TFs, as many of these TFs, including Fer1, Sug, Dac, Poxn and Hbn, were identified by Pros-DamID as positive hits (Fig. 2d). To further validate a direct role for Pros in the regulation of *fer1* transcription, we generated a Fer1$^{Pros}$-lacZ reporter using the identified putative Pros binding region (Supplementary Fig. 8a). Fer1$^{Pros}$-lacZ marked a subset of EEs located at R4 region (Supplementary Fig. 8b, yellow arrow heads), and its expression was entirely abolished following *pros* depletion (Supplementary Fig. 8c, white arrow heads). Therefore, Pros may directly regulate the transcription of *fer1*. These observations support a direct role for Pros on the transcription of both 2⁰ (Fig. 2m–r) and 3⁰ TFs (Supplementary Fig. 6a–c).

We next tested whether the transcription of 3⁰ TFs is dependent on 2⁰ TFs. By RT-PCR analysis of gut extract, we found that conditionally knocking down the 2⁰ TF *ptx1* or *mirr* in EEs driven by ProsV1-Gal4$^{ts}$ for 7 days did not significantly affect the expression level of 3⁰ TFs (Supplementary Fig. 9a, b), indicating that the transcription of 3⁰ TFs is not dependent on 2⁰ TFs. As depletion of either *ptx1* or *mirr* in EEs abolishes the expression of all class I or class II subtype- specific neuropeptides, respectively[10], Ptx1 or Mirr is likely continuously

required as a class level TF, which functions together with 3⁰ TFs to further define subclass EE subtypes.

As described earlier in this study, we constructed two lacZ reporters driven by the Pros binding regions on NPF and CCHa1 loci. NPF expression is normally restricted to several EE subtypes within the class II group, including II-a, II-m1, II-m2 and a part of II-p[10]. We found that this lacZ reporter exhibited spatial distributions along the midgut in a similar way as the native expression patterns of the corresponding neuropeptide: the NPF$^{Pros}$-lacZ positive cells are most abundant in R3 (middle midgut) and to a less extent R2 (anterior midgut) EEs. In addition, co-staining of the lacZ reporter with the corresponding gene knock-in reporter line NPF-LexA>GFP revealed perfect overlapping patterns (Supplementary Fig. 10, yellow arrowheads), showing that the enhancer reporter line can faithfully reflect the transcriptional activity of the corresponding genes, which also implies that the NPF$^{pros}$ enhancer region used to drive the reporter expression is able to recruit additional TFs for the specification of subclass EE subtypes. To further test this hypothesis, we examined the effect of previously-identified NPF-regulating TFs on the expression of the NPF$^{pros}$-lacZ reporter. The 2⁰ TF Mirr and 3⁰ TF Esg are two positive regulators of NPF[10], with Esg as a regional TF, because it is only expressed in EEs at R3 region[10,29]. As expected for a role of Mirr in specifying class II subtype identity, knocking down *mirr* in EE cells driven by ProsV1-Gal4$^{ts}$ for 7 days abolished NPF$^{Pros}$-lacZ expression in all of the presumptive class II subtypes along the length of the midgut (Fig. 5a, b). Conversely, overexpression of *mirr* induced NPF$^{Pros}$-lacZ signal in virtually all EEs (Fig. 5a–e). These observations not only are in consistent with Mirr as a master regulator of class II subtype identity, but also suggest that Pros and Mirr act cooperatively on a common enhancer region to promote the transcription of their target genes. Similarly, knocking down *esg* for 7 days specifically downregulated NPF$^{Pros}$-lacZ expression in EEs at R3, but not at other regions (Fig. 5f). Therefore, Esg also functions through a common enhancer region with Pros to regulate NPF expression in R3 EEs.

The neuropeptide CCHa1 is expressed in several subtypes within both class I and II groups, including I-ap, I-p, II-p and a small part of II-a[10]. The expression of CCHa1$^{Pros}$-lacZ, however, was found only in a subset of CCHa1-LexA>GFP$^{+}$ cells (Supplementary Fig. 11). In addition, CCHa1$^{Pros}$-lacZ was only expressed in AstC$^{+}$ EEs but not TK$^{+}$ EEs (Fig. 5g, h), indicating that this enhancer region lacks certain element required for the expression in class II subtypes. Nevertheless, we tested the ability of the 2⁰ TFs in the regulation of CCHa1$^{Pros}$-lacZ expression. We found that knocking down *ptx1* caused loss of CCHa1$^{Pros}$-lacZ expression in all EEs, thus confirming the requirement of Ptx1 for CCHa1 expression in class I subtypes (Fig. 5i). Interestingly, depleting *mirr* allowed CCHa1$^{Pros}$-lacZ expression in almost all EEs (Fig. 5j, yellow arrowheads), while *mirr* overexpression in EEs led to loss of both CCHa1$^{Pros}$-lacZ and AstC expression (Fig. 5k). These observations are not entirely surprising, as loss of *mirr* is able to cause class II to class I subtype switch, which allows Ptx1-mediated specification of class I subtype identity and consequent expression of the CCHa1$^{Pros}$-lacZ reporter. Therefore, the expression of CCHa1 in class I subtypes also simultaneously requires both Pros and Ptx1 on a common enhancer region, further supporting the notion that 1⁰ and 2⁰ TFs act cooperatively to specify the two classes of EE subtypes, thereby enabling appropriate neuropeptide expression patterns.

Fer1 is a 3⁰ TF expressed in a subset of CCHa1$^{+}$ EEs, and is a potential regulator of CCHa1 expression[10]. Indeed, knocking down *fer1* in EEs diminished CCHa1$^{Pros}$-lacZ expression in AstC$^{+}$ EEs without affecting AstC or Tk expression patterns (Fig. 5l and Supplementary Fig. 12a, b). These results suggest that the 3⁰ TF Fer1 functions with Pros and Ptx1 to further define CCHa1 transcription in a subset of class I subtypes, thereby contributing to the establishment of sub-class EE subtype identities.

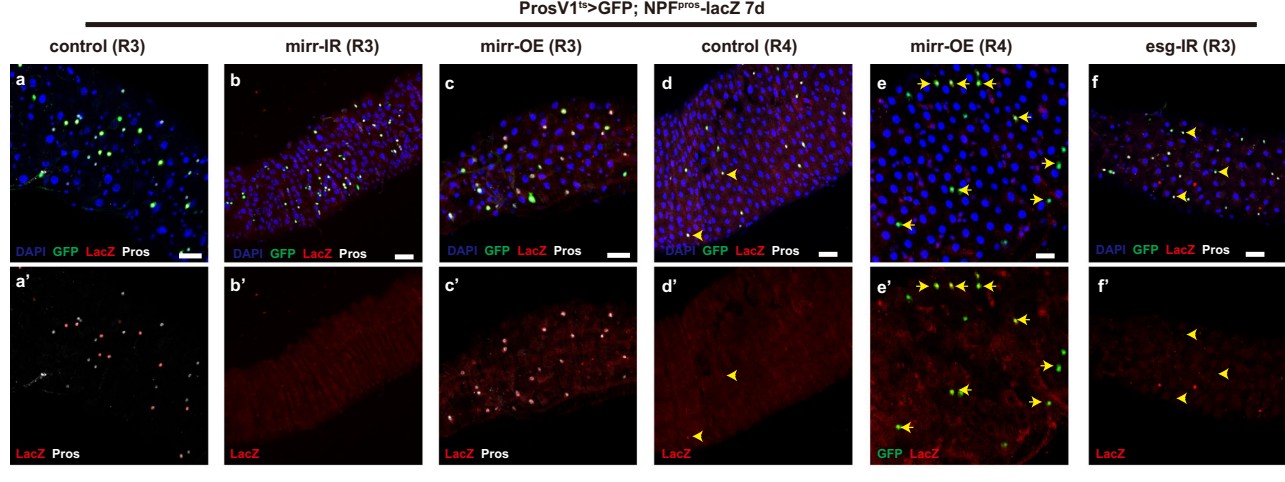

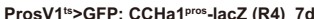

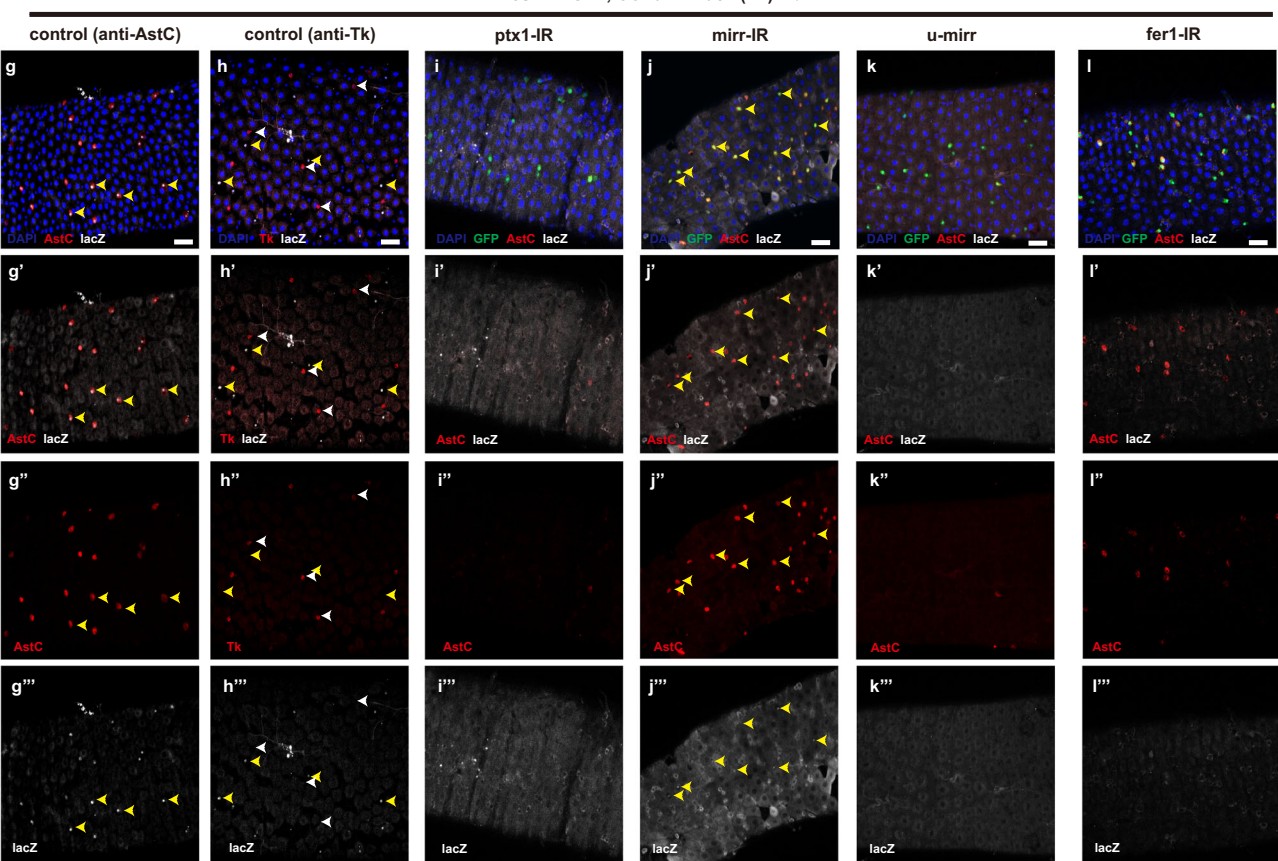

**Fig. 5 | Pros and 2⁰ TFs cooperate with 3⁰ TFs to define subclass EE subtypes.**
**a–f** Expression of NPF$^{Pros}$-lacZ in both R3 (**a**) and R4 (**d**) regions and its regulation by Mirr (2⁰ TF) and Esg (3⁰ TF), a region-specific EE factor. Knocking down *mirr* (**b**) and *esg* (**e**) by ProsV1-Gal4$^{ts}$ significantly down-regulates NPF$^{Pros}$-lacZ level, while over-expressing *mirr* turns on NPF$^{Pros}$-lacZ expression in almost all EEs (**c**, **e**). Knocking down *esg* suppresses NPF$^{Pros}$-lacZ expression in R3 EEs (**f**); co-staining of CCHa1$^{Pros}$-lacZ with AstC and Tk reveals that CCHa1$^{Pros}$-lacZ expression is expressed in AstC⁺ EEs (**g**, tallow arrowheads), but not in Tk⁺ EEs (**h**, white arrowheads); **i–l** Expression of CCHa1$^{Pros}$-lacZ is controlled by both 2⁰ and 3⁰ TFs. Knocking down 2⁰ TF Ptx1 disrupts both AstC and lacZ expression (**i**); knocking down 2⁰ TF *mirr* turns on both AstC and lacZ expression in the majority of EEs (**j**); *mirr* overexpression suppresses both AstC and lacZ expression (**k**). Knocking down the CCHa1- regulating TF Fer1 (3⁰ TF) significantly downregulates CCHa1$^{Pros}$-lacZ expression in EEs (**l**).

Collectively, our results indicates that, in addition to the cooperative relationships among 1⁰, 2⁰ and 3⁰ TFs in determining EE subtype identities, there are two aspects of hierarchical relationships among 1⁰, 2⁰ and 3⁰ TFs in the process: the 1⁰ TF (Pros) acts at the top of the hierarchy to regulates both 2⁰ and 3⁰ TFs; and there is also a functional hierarchy in which the function of 2⁰ TF is dependent on 1⁰ TF, but not vice versa, and the function of 3⁰ TFs is dependent on both 1⁰ and 2⁰ TFs, but not vice versa.

## Fer1 is a determinant of subclass EE subtypes

To further validate whether Fer1 directly regulates the transcription of CCHa1, we conducted Fer1 Dam-ID analysis to determine its putative

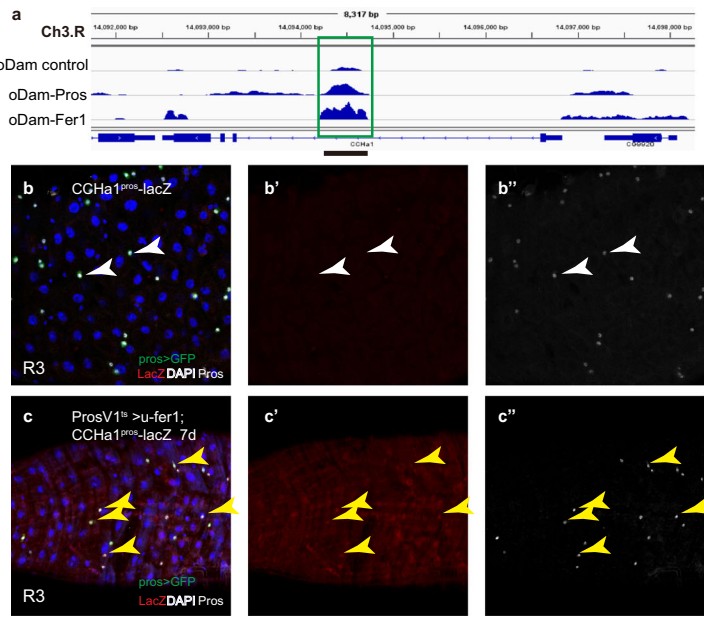

**Fig. 6 | The 3° TF Fer1 is a determinant of sub-class EE subtypes. a** The putative Pros targeting region on CCHa1 loci (also the region used to generate CCHa1^pros^-lacZ reporter) is also targeted by Fer1, as indicated by the significant peak of oDam-Fer1 on the underlined region; **b, c** the CCHa1^pros^-lacZ reporter is responsive to ectopic Fer1 activation. In normal guts, CCHa1^pros^-lacZ reporter is not expressed in R3 EEs (**b**, white arrowheads), while Fer1 overexpression turns on lacZ expression in R3 EEs (**c**, yellow arrowheads). In normal guts, CCHa1 expression is absent in R3 region (**d**), while Fer1 overexpression in EEs robustly induces CCHa1 expression in R3 EEs (**e**). Scale bars, 50 μm.

targeting sites. As shown in Fig. 6a, a Fer1 targeting region on CCHa1 was identified, and this region is perfectly co-localized with the CCHa1^pros^ region. This observation reinforces the idea that this enhancer region recruits both Pros and Fer1 to regulate the transcription of CCHa1.

CCHa1 is normally not expressed in EEs at R3 region (Fig. 6d). Interestingly, revealed by a CCHa1-lexA knock-in reporter, we found that ectopic expression of *fer1* for 7 days was able to induce CCHa1 expression in EEs at R3 (Fig. 6e). Similarly, ectopic expression of Fer1 also ectopically induced expression of CCHa1^pros^-LacZ in R3 EEs (Fig. 6c), where it is normally not expressed (Fig. 6b). Thus, Fer1 is both necessary and sufficient in determining CCHa1 expression in subclass EE subtypes, and Fer1 can be considered as a determinant of subclass EE subtypes. These results further demonstrate an impressive degree of cellular plasticity of EEs that allows adoption of different subtype identities and neuropeptide profiles by simply altering the expression of a single TF.

## Discussion

Understanding cellular diversity and the underlying mechanisms is an important issue in cell and developmental biology. For well-appreciated examples, the immunological T and B cells use V(D)J recombination to generate diverse repertoires of T and B cells[30]; The diversity of olfactory neurons relies on chromatin architecture regulation of the olfactory receptor locus that allows random expression of a distinct spliced form of receptor in each cell[31]; and the diversification of neurons in the nervous system is controlled by a collection of TFs, with unique combination of TFs that define certain subtypes. Importantly, these TFs exhibit both temporal and spatial changes in their expression patterns and functional requirements[32,33]. Here, we identify a mechanism in controlling the cellular diversity of *Drosophila* EEs that is distinct from any of the above mentioned models, which involves cooperative action of multiple TFs at multiple layers in a regulatory hierarchy: (1) The homeodomain TF Pros serves at the top of hierarchy for the entire EE identity by globally regulating the transcription of both the second level (2°) TFs including Ptx1 and Mirr, and the third level (3°) TFs, which includes local or regional factors; (2) The combination of Pros and the 2° TFs determines the identity of the two major inter-switchable subtype groups (classes I and II); (3) The combination of Pros, 2° TF, and 3° TFs further refines enhancer activity and determines subclasses of EE identity. As the expression of 2° TFs are also regulated by Notch signaling, and 3° TFs by local (position information resulted from body patterning) and environmental factors, the three part code thus can create considerable code diversity, thereby enabling locally and spatially diverse neuropeptide expression patterns along the *Drosophila* midgut (Fig. 7).

A gene with an ability to determine the identity of a specific organ, tissue, or cell type is referred to as a selector gene[34]. Our study suggests that *pros* is a selector gene for *Drosophila* EEs, as it directly orchestrates the EE-specific transcriptional programs and it is responsible for maintaining the identity of the entire EE cell population. Is there a selector gene for mammalian EEs? There are a number of TFs implicated in the regulation of EEs or EE subtypes in the mammalian digestive tract[1,5]. The bHLH family TF Neurog3 is considered to be at the top of the regulatory network and is required for the specification of the entire EE subtypes[35]. However, Neurog3 expression is diminished as EEs mature, implying that this TF is dispensable for maintaining mature EE identity[36]. Interestingly, Prox1, the mammalian ortholog of Pros, is expressed in certain EE subtypes expressing PYY, cholecytokinin (CCK), and GLP-1, while serotonin-expressing enterochromaffin cells are rarely Prox1^+[37,38]. Thus, Prox1 is unlikely the major TF for the specification and maintenance of the EE identity in mammals, and the selector gene for mammalian EEs, if there is one, remains to be identified.

Similar to *Drosophila*, the mammalian EEs also diverge into two major subtype groups (enterochromaffin cells and non-enterochromaffin cells) during their specification from progenitor cells, and many subtype specifically-expressed TFs are also evolutionarily conserved between *Drosophila* and mammals[5]. In particular, IRX3, the ortholog of *Drosophila* Mirr, is specifically expressed in enterochromaffin cells. This EE subtype also specifically expresses *Tachykinin precursor 1* (*TAC1*), which belongs to the Tachykinin family genes[39]. Thus, although the role of IRX3 in mammalian EEs remains to be characterized, it appears that the *Drosophila* class II EE subtype is

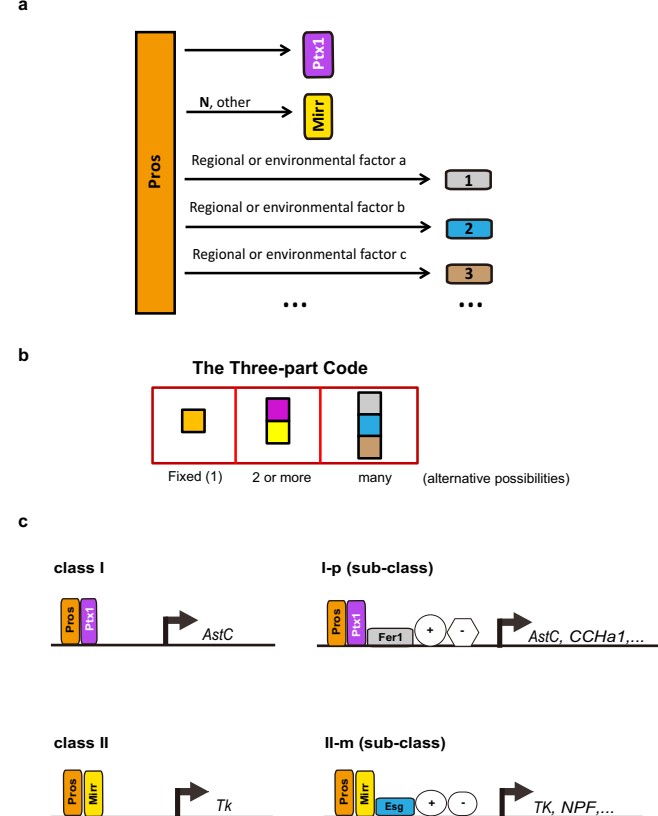

**Fig. 7 | The hierarchical model and the three-part code principle underlying EE subtype diversity. a** Pros is a selector protein in defining EE identity, as it serves at the top of the hierarchy and globally regulates the transcription of both the second level ($2^0$) TFs [including but not limited to Ptx1 and Mirr (whose expression also requires Notch activity)], and the third level ($3^0$) TFs, the expression of many of which also requires local or regional factors. **b** The combination principles of the three-part TF code and the theoretical combination possibilities underlying EE diversity. There is only one $1^0$ TF (Pros); the $2^0$ TFs include Ptx1 and Mirr, and potentially additional ones, such as the ones for class III EE identity; and there are multiple (>12) $3^0$ TFs. **c** The TF code model in determining class EE subtypes and sub-class EE subtypes. The combination of Pros and one of the $2^0$ TFs determines the identity of the major subtypes (such as class I and II); and the combination of Pros, $2^0$ TF, and $3^0$ TFs further refines enhancer activity and determines subclass EE subtypes and hormone expression profiles. Conceivably, further addition of co-activators or co-repressors may allow additional diversity of EE subtypes, and/or generation of dynamic hormone expression patterns for each specific EE subtypes.

probably analogous to the mammalian enterochromaffin cells in its biological function and TF regulation. The recent single cell level analyses in mammalian and human intestine have revealed TFs that are expressed in the entire EE cells as well as TFs expressed in specific EE subtypes[36,40], which should help to guide the identification of general and subtype-specific EE regulators and test the three part code hypothesis.

Another notable finding from this study is that the differentiated EEs still show a high degree of post-specification plasticity and are amenable to subtype identity changes. EE cell plasticity has been also observed in mammals, in which the L cells located near the crypt bottom switch their hormone expression profile and subsequently become cholecystokinin-expressing I cells and neurotensin-expressing N cells as they migrate up toward the tips of villi[36,41]. Another study in mice showed that somatic ablation of FOXO causes EEs of the small intestine to become beta-cell like cells and express insulin[42]. Therefore, in addition to the similarities in cell lineage hierarchy and the shared TF regulators, the post-specification plasticity is also a common feature of EEs in both *Drosophila* and mammals. We believe that the ideas of the

selector gene and the hierarchically-organized three part code reported here can serve as a predictive conceptual framework to facilitate investigations of EE cell specification, maintenance, and plasticity in mammalian systems, and along with our continuously-increased understandings of subtype-specific biological functions of EEs and neuropeptides, genetic or pharmacological manipulating of EE subtypes could bring promising clinical values toward diabetes, obesity, and other endocrine-related diseases.

## Methods

### Fly strains and cultivation

The following fly strains were used in this study: prosV1-Gal4, UAS-GFP (gift from Bruce Edgar); Rab3-EYFP (BDSC, #62541); UAS-RFP, LexAop-GFP (BDSC, #32229); CCHa1 -R-LexA; CCHa1-LexA; NPF-LexA; AstA-LexA; CG32547-Gal4; DH31-LexA; AstC-Gal4; Tk-Gal4; UAS-oDam; these lines were previously generated in our lab. UAS-oDam-Pros (This study); mirr[B1-B12]-lacZ (BDSC, #10880); CCHa1[Pros]-lacZ (This study); CCHa1[Pros Δmotif]-lacZ (This study); NPF[Pros]-lacZ (This study); NPF[Pros Δmotifs]-lacZ (This study); Ptx1[Pros]-lacZ (This study); Fer1[Pros]-lacZ (This study); myo1A-Gal4, UAS-GFP; esg-Gal4, UAS-GFP; UAS-*Pros* (BDSC, #32244); UAS-*Ptx1* (generated in this study); UAS-*Fer1-3\*HA* (FlyORF, F000074); UAS-*mirr-3\*HA* (FlyORF, F001890); UAS-*Nicd* (gift from Ting Xie); UAS-*pros-RNAi* (JF02308, Chr.3; HMJ02107, Chr.2); *mirr-RNAi* (JF02196, Chr.3; SH05171.N, Chr.2); *fer1-RNAi* (JF02821, Chr. 3); *Ptx1-RNAi* (SH05053.N, Chr.3); *esg-RNAi* (HMS00025, Chr.3); *Notch-RNAi* (BDSC, #7078); *UAS-Flp*, *Act < stop < lacZ* and *Tub-Gal80[ts]* were all obtained from BDSC. Fly stocks were cultivated on standard food with yeast paste added on the food surface and kept at 25 °C unless otherwise stated.

The Gal4/UAS/Gal80[ts] and LexA-lexAop systems were used to conduct conditional knocking down or overexpression in specific cell types. Unless otherwise stated, all crosses were performed at 18 °C, and 3–7 day old adult F1 progenies with correct genotype would be collected and transferred to 29 °C to induce gene expression.

### Generation of transgenic flies

**Generation of pUAST-oDam-Pros transgenic fly.** Whole length of *pros* CDS was PCR amplified and inserted into the multiple cloning site of the previously generated attB-pUAST-oDam vector[20]. After sequencing verification, the transgene were then inserted into attP2 sites using the phiC31 system via a standard microinjection process.

**Generation of the CCHa1[Pros]-lacZ, NPF[Pros]-lacZ, Ptx1[Pros]-lacZ and Fer1[Pros]-lacZ transgenic flies.** The putative binding regions of Pros on CCHa1 (+1703 bp to +2218 bp of the TSS), NPF loci (−2064 bp to −1310 bp of the TSS), Ptx1 loci (+2432 bp to +3619 bp of the TSS) and Fer1 loci (−469bp to +634 bp of the TSS). After PCR amplification of these regulatory elements, they were enzymatically digested and separately inserted into the multiple cloning sites of the C4pLZ vector (RRID:DGRC_1041). After sequencing verification, these transgenes were subsequently injected to w1118 fly embryos via a standard microinjection process. The reporters were randomly integrated into fly genome by P-element insertion.

**Generation of the Pros binding motif depleted CCHa1[Pros Δmotif]-lacZ, NPF[Pros Δmotif]-lacZ, transgenic flies.** The CCHa1[Pros]-lacZ and NPF[Pros]-lacZ vectors described above were used as templates, and the Quick-change method was used to delete the predicted Pros binding motifs within oDam-Pros binding sequences on CCHa1 and NPF loci. Dpn I was subsequently used to digest CCHa1[Pros]-lacZ and NPF[Pros]-lacZ vectors, while motif depleted CCHa1[ProsΔmotif]-lacZ, NPF[ProsΔmotif]-lacZ vectors were transfected into DH5α competent cell. After sequencing verification, these two mutated transgenes were injected to the w[1118] fly embryos via a standard microinjection process.

## Immunostaining

Immunostaining of *Drosophila* midgut was performed as previously described[43]. In brief, 10–15 adult female flies for each sample were dissected in ice-cold 1XPBS and then fixed in 4% paraformaldehyde for 30 min at room temperature, followed by dehydration in methanol (for 5 min) and rehydration in PBT solution (PBS containing 0.1% Triton X-100, 5 min each for three times). The primary antibodies were added into 300 μl 5% NGS-PBT solution and incubate with sample for 2 h at room temperature or overnight in 4 °C. After washing three times using PBT, the secondary antibodies were added into 300 μl PBT and incubate with samples for 2 h at room temperature, followed by staining of DAPI for 5 min. In total, 70% glycerol was used to mount the samples and slides were kept at −20 °C freezer. Images were captured using confocal microscope systems. The classification of R1-R5 regions along the anterior and posterior midgut is as previously described[44], and the gut images shown in the figures are from the R4 region by default, unless otherwise noted.

Primary antibodies used in this study were listed as follows: mouse anti-Pros (DSHB #MR1A; 1:300); rabbit anti-AstC (lab generated antibody (RRID: AB_2753141) and gift from Dr. Dick Nassel; 1:300); rabbit anti-Tk (lab generated antibody (RRID: AB_2569591) and gift from Dr. Jan-Adrianus Veenstra; 1:300); rabbit polyclonal anti-β-galactosidase (Cappel, 0855976; 1:6000); rabbit polyclonal anti-Cleaved Drosophila Dcp-1 antibody (AB_2721060; 1:300); mouse monoclonal anti-β-galactosidase (DSHB,# 40-1a; 1:30). Secondary antibodies used in this study include Alexa Fluor 488-, 568- or Cy5-conjugated goat anti-rabbit, anti-mouse IgGs (Molecular Probes, A11034-A11036, A10524; 1:300), and DAPI (Sigma-Aldrich, 1 μg/ml) was used for nuclei staining.

## Fluorescence-activated cell sorting (FACS)

Cell sorting was performed following previously reported protocols[10,45]. In brief, 100–150 guts for each sample (3 independent replicates for each genotype) were dissected in ice-cold DEPC-PBS within 2 h, and were digested in 1 mg/ml elastase solution (Sigma, cat. no. E0258) for 1 h at room temperature with gentle shakes. The dissociated samples were then centrifuged at $500 \times g$ for 10 min at 4 °C, and re-suspended in 500 μl DEPC-PBS with 1 mg/ml propidium iodide (PI, Invitrogen, #P3566). Around 20,000 PI$^-$ GFP$^+$ cells were collected for each sample using a FACS Aria II sorter (BD Biosciences) with FACSDiva software (Version 6.1.3), following the gating strategy for FACS sorting as displayed in Supplementary Fig. 13 (the cells in P5 were sorted and used for subsequent RNA-sequencing).

## RNA-sequencing and data analysis

Five to seven day old adult female flies with the following genotypes were cultivated in 29 °C for 7 days and been used for RNA-sequencing: prosV1-Gal4$^{ts}$, UAS-GFP and prosV1-Gal4$^{ts}$,UAS-GFP > UAS-*pros-RNAi*. Dissection, tissue digestion and FACS sorting were carried out following the procedure described above. For each sample, around 20,000 PI$^-$ GFP$^+$ cells were FACS sorted into 300 μl RNA extraction buffer. The following RNA extraction and amplification steps were carried out using Arcturus PicoPure RNA isolation kit (Applied Biosystems, Cat#KIT0204) and Arcturus RiboAmp HS PLUS RNA amplification kit (Applied Biosystems, Cat#KIT0525) respectively, following the manufacturer's instructions.

The amplified RNA were used for subsequent sequencing, following the protocols previously reported[10,20]. Briefly, the NEB Next Ultra II DNA library prep kits (New England Biolabs, cat. no. E7645L) was used for library preparation, and single ended deep sequencing was carried out on an Illumina Hiseq-2500 sequencing system with 50 bp read length. Raw reads were mapped to D. melanogaster genome (BDGP6) and counts assigned to protein-coding genes were calculated using featureCounts (v1.6.3). DESeq2 was then used to identify significantly differently expressed genes using the following

parameters: $P$adj < 0.01, and the absolute value of log2 FC > 0.5. GO analysis for differently expressed genes was performed using DAVID[46], and the R package "pheatmap" was used for generating heatmaps.

Bulk RNA-sequencing profiles for EE (labeled by ProsV1-Gal4), progenitor (labeled by esg-Gal4) and EC cells (labeled by Myo31DF-Gal4), were also used for choosing cell type marker genes. For each cell type, low abundantly expressed genes with RPKM value less than 3.5 were removed, and each gene was scored by using following formular: Fold change = (RPKM in one cell type)/(mean RPKM in all three cell types); Score = log10(RPKM)/5 + Fold change. Finally, the top 250 scored genes were assigned as cell type markers. These marker genes were further used as gene sets for GSEA analysis.

## Dam-ID sequencing and data analysis

Adult female flies with the following genotypes were cultivated in 29 °C for 48 h before gut dissection and Dam-ID analysis following the protocol modified from previous reports[24]: prosV1-Gal4$^{ts}$,UAS-GFP; UAS-oDam and prosV1-Gal4$^{ts}$,UAS-GFP; UAS-*oDam-Pros*. In brief, 50–60 midguts were dissected for each sample, and total genome DNA was extracted, followed by DpnII digestion, dephosphorylation (twice the amount of enzymes and treatment time compared to the original protocol), DpnI digestion and T4 ligation of adapters. Afterwards, fragments were amplified by ligation-mediated PCR, T7 exonuclease treatment and purification, yielding DNA products ranging from 200 to 2000 bp. Finally, the products were fragmented and carrying out subsequent library preparation and high-throughput sequencing.

Raw sequencing reads were mapped to the *D. melanogaster* genome (BDGP6) using bowtie2 (version 2.2.4). R package iDEAR 0.1.0 was subsequently used for establishing highly reliable profiles of Pros-binding sites in *Drosophila* EEs. The density of Pros-binding on each genomic region was normalized to the total number of mapped reads. BigWig files were generated for visualization using the Homer package. Annotated oDam peaks (oDam-Pros versus oDam) with log2Fold-Change more than 1 and adjusted $p$ value <0.01 were filtered as putative Pros-binding targets.

## Combined analysis of RNA-seq and Dam-ID results

To further filter genes directly targeted by Pros, we carried out combined analysis of RNA-seq and Dam-ID results. Genes exhibiting both significant binding by Pros in Dam-ID analysis and up or down-regulated expression level in *pros*-depleted EE cells were selected. The top 250 marker gene panels described above were further used to perform GSEA analysis, to evaluate whether identity genes of certain cell type was globally targeted by Pros and the effects of *pros*-depletion on their expression in EEs. The GSEA analysis was performed done using the R package "clusterprofiler".

## Statistics and reproducibility

All experiments were reproduced independently for at least 2–3 times with similar results, and representative results were shown in the manuscript.

Cell number counting was performed using Image J (1.48v), and all quantifications were presented in the form of mean ± SEM. GraphPad Prism 5 software (GraphPad Software Inc.) was used to calculate $p$ values by unpaired Student's $t$ test or ANOVA test.

## Quantitative PCR (qPCR) analysis of subtype regulatory TFs expression

Tub-Gal80$^{ts}$; ProsV1-Gal4, UAS-GFP strain was used to specifically knock down Mirror or Ptx1 in EEs. Crosses were performed at 18 °C to avoid premature expression. Five to seven day old adult F1 females with correct genotypes were collected and cultivated at 29 °C for 5 days. Total RNA extraction, cDNA synthesis, and qPCR analysis were carried out on the Applied Biosystems 7500 Real-Time PCR System as previously described[10,20]. Expression levels of TFs were normalized to

*rp49*, and qPCR primers used in this study were listed in Supplementary Data 7.

### Reporting summary

Further information on research design is available in the Nature Research Reporting Summary linked to this article.

## Data availability

The data that support this study are available from the corresponding author upon reasonable request. The raw and processed datasets, including RNA-seq data and Dam-ID results generated in this study have been deposited in the GEO database under accession code GSE211632 Two RNA-seq datasets previously reported by our lab have also been used in this study and were available in the GEO database, under the accession code GSE130943 (RNA-seq data of esg+ cell) and GSE130305 (RNA-seq data of EC cell). Source data are provided with this paper.

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

## Acknowledgements

We thank Dr. Dick Nassel (Stockholm University) for providing anti-Tk antibody, Dr. Jan-Adrianus Veenstra (Université de Bordeaux) for anti-AstC antibody, and the Bloomington *Drosophila* Stock Center (BDSC), the Tsinghua Fly Center, and Development Studies Hybridoma Bank (DSHB) for fly strains and antibodies, and members of the Xi laboratory especially Jiaying Lv and Diyi Yang for discussion and reading of the manuscript. This work was supported by National Key Research and Development Program of China (2020YFA0803502 and 2017YFA0103602 to R.X.) from the Chinese Ministry of Science and Technology. X.G. is supported by National Natural Science Foundation of China (Grant No. 3210050518).

## Author contributions

X.G. and R.X. designed the experiments. X.G. performed the experiments. Y.Z. and H.H. performed the informatics analysis. R.X. acquired funding. X.G., Y.Z. and R.X. wrote the paper.

## Competing interests

The authors declare no competing interests.
