## [Peer Review File · Nature Communications]

REVIEWER COMMENTS

Reviewer #1 (Remarks to the Author):

This paper from Guo, Xi, and colleagues extends the current understanding of enteroendocrine (EE) cell specification in the *Drosophila* gut by focussing on the actions and interactions of a set of transcription factors that determine how ~10 different EE identities are established as the progeny of EE progenitors differentiate. The genetic analysis used is similar to what was done for fly neural stem cells more than a decade ago, and the general result is not dissimilar: EE progeny are subject to hierarchical regulation by different combinations of transcription factors (TFs) that determine specificity at higher or lower levels. This is not unexpected, and indeed several publications already define the different EE subtypes (by hormone expression combinations) and report some details of their specification. However this current analysis adds many details, including data on new TFs, and illustrates the logic of specification in a much more comprehensive way. It's a good quality analysis and overall I liked it very much. It will provide a useful framework to guide future studies of EE specification in *Drosophila* as well as mice and humans. I have a number of relatively minor comments the authors should address as they revise the paper, most of which pertain to adding details such that the description of the analysis is more accurate and complete.

1. In most experiments a temperature sensitive Gal80-ts gene is used to control the time of RNAi or gene target activation. Please add details of when the TS-Gal80 was used, and when not, and also provide the timing of temperature shift and subsequent incubation for each experiment. This info can be denoted in the figures and figure legends.

2. The text has many grammatical mistakes, some of which obscure meaning. These should be corrected.

3. The Results section often mentions information from other publications without citing them. Please make sure all references to perviously published data include citations.

4. On page 8 line 4, and Fig 2D "pros target genes" are described. But it is not clear how this subset of genes was defined. Please give a precise description of out this pros target gene subset was defined.

5. The reporter genes described first on page 8 (2nd paragraph) and in Fig 2H-M are very nice. However the authors should describe these more precisely and provide maps of them.

6. It could be very informative to make versions of these reporter genes with the putative pros binding sites mutated, and analyze their expression in various situations.

7. Although it is known that different EE subtypes have distinct regional distributions in the fly midgut, regional variations in EE type and specification mechanisms are rarely mentioned in this paper. The authors should try to include this information. At least, they should indicate which regions were assayed to reach the conclusions that they make. EE specification mechanisms are probably different in different midgut regions, and the authors do not acknowledge this issue sufficiently.

8. On page 15, the authors describe how the Fer1 TF affects EE specification and the expression of their CCHA1-pros-LacZ reporter. This is quite interesting, but the results would be even better if they were able to predict the Fer1 binding site in the CCHA1 target gene, and test their prediction by mutating the site(s) in their CCHA1 reporter.

9. The paper very much needs a summary figure that clearly lays out the hierarchy of factors that control the different EE identities. Such a figure will help guide future research on this topic.

Reviewer #2 (Remarks to the Author):

Guo et al present an interesting study on enteroendocrine cell specification and plasticity. They examine the role for Prospero in EE specification and identity maintenance and discover presumptive Pros downstream genes and binding sites to include both the two class specific TFs as well as some subtype specific TFs. Further, they explore the post differentiation plasticity of the EEs by depleting or overexpressing class specific TFs.

While overall the data is compelling and worth publishing, additional experiments/controls need to be performed to justify the conclusions. This is especially true regarding Prospero's role in EE identity maintenance and expression of subtype specific genes. Moreover, the conclusion that the authors make that there is a top-down hierarchical TF cascade doesn't fit their data. While Pros seem likely to control the 2nd tier TF (with notch binary sister fate), the 2nd tier do not directly control 3rd tier TFs. In particular, subtypes reflect regional identity, suggesting a combinatorial mechanism (rather than hierarchical).

Major points:

1) Much of the methodology is not stated, making it difficult to understand and interpret the data.

a. For example, in Figure 1A-G) there is no explanation of how the experiments were performed.

i. How long is the temperature shifted before analysis?

ii. What stage is analyzed?

iii. How is Pros depleted? Presumably UAS-Pros-RNAi, but it is not stated in the results, legends or methods.

2) The conclusions about Pros maintaining EE identity are not convincing.

a. prosV1Gal4ts is a problematic reagent to deplete Pros only from mature cells. 1)Pros is expressed starting from an immature stage. 2)Pros depletion may affect the expression of the prosV1Gal4ts.

i. Are there other reagents to specifically target the mature EE state?

b. While figure 1A-G methods are not described, based on the RNA sequencing methods section, it seems likely that Pros depletion was performed via a 7-day temperature shift at 29 degrees (expressing pros-RNAi). If this is the case, it is difficult to believe that the same "mature" EE cells can be visualized 7 days later. It is very possible the GFP+ cells are actually born post Pros depletion-making them immature EE cells. The ISC division rate and EE to ISC ratio (Marianes and Spradling) suggest that many (if not most) GFP+ cells after 7days at 29 could have been born after initiation of Pros depletion.

i. The time course in Figure S1 doesn't really address these concerns as it is presented. Rather than "a gradual loss of Pros expression... [ac]companied by a gradual loss of hormone expression," a gradual loss of Pros+ and Astc+ cells is observed. This could easily be explained by cellular turnover (cell death/increase of new EEs).

c. In the time course (FigS1) do the number of GFP+ cells change 1-3d after temperature shift (when a significant decrease in AstC+ cell numbers is seen)?

d. Apoptotic markers should be examined.

e. If 1d post temp shift is sufficient for a significant decrease in AstC expression, can peptide hormone expression also recover in a similarly short time-period?

3) The same issues regarding the identity of the cells (as mature EEs lacking Pros) being analyzed also apply to the FACS sorted cells. Perhaps even more so, as these are 5-7day old flies that have been shifted to 29 for 7 days, suggesting that Pros has been depleted since prior to eclosion and thus completion of adult gut development.

a. How does the interpretation of the data change if the cells analyzed are immature EEs, not yet expressing Pros?

4) The Dam-ID (Pros-Dam) is directly driven by prosV1Gal4ts, making it highly expressed compared to traditional Dam-ID or TaDA which require very low expression levels to prevent oversaturation and potential binding artifacts.

a. Additional evidence needs to be provided that the EE TFs are Pros target genes (particularly those with not as significantly enriched binding peaks, like Ptx1).

b. How is Pros knocked down in Figure 2F,I&L? could loss of LacZ be due to change in cell fate?

c. Does Pros overexpression cause increase in LacZ levels?

5) Introduction is lacking—in particular:

a. the nature of EE diversity (explain the neuropeptide hormones better)

b. what is already known about the TF code of the different types of EE cells (work from their lab).

i. How are they defining tertiary TFs?

c. The role of Notch in class 1 v class 2-what is already known.

6) FIGURE 3

a. The “cell lineage” experiments are not actually lineage tracing because they use dynamic GAL4 drivers.

b. Figure 3A-F and 3J-M, single-color images should be shown.

c. Again, details are not given about how long genes are depleted or overexpressed is it 7 days?

d. there seems to be cell loss (see below)

i. In Mirr-IR see lots of EE doublets expressing AstC, but with UAS-mirr fewer doublets are obvious and those clear doublets appear to have one cell expressing TK and the other not....Are there the same # of GFP+ cells with UAS-mirr as control or mirr-IR? They look like they may be fewer in number (zoomed out a little too, so perhaps UAS-mirr causes death?).

ii. 3K (UAS-mirr) also appears to be missing doublets compared to 3J are there fewer EEs in 3K?

iii. There are clearly fewer EEs in 3M and no doublets.

7) FIGURE 4

a. Again more methods needed

b. For 4I-O is a single Pros downstream gene enough to call it transdifferentiation? Do other not immediate Pros downstream EE specific genes need to be turned on to say “transdifferentiation?”

8) FIGURE 5

a. First paragraph of “Pros cooperates with 2...” is very confusing.

b. The text says “As described previously, Pros directly regulated the transcription of both secondary and tertiary TFs,” yet the tertiary TFs have not been fully described or defined. There is not definitive evidence that Pros directly regulates tertiary TFs.

Reviewer #3 (Remarks to the Author):

This is an interesting follow-up study to a study published in Cell Reports by the same authors in which the authors had used scRNAseq to identify a transcription factor code that presumably controls enteroendocrine cell diversity in the Drosophila midgut. Here the authors test this hypothesis by performing a series of genetic perturbations combined with RNAseq and DamID that reveal TF dependencies in subtype maintenance in the enteroendocrine lineage.

Some of the findings reported here have been reported in the previous study (such as that Mirr knockdown inhibits Tk expression while Ptx1 knockdown inhibits AstC expression), but other convincingly make the case that subtype identities are maintained by combinatorial gene regulation by Pros, Ptx1/Mirr, Esg/Fer. As such it is an interesting study reporting convincing results from well-performed experiments. A major editorial question is whether the advance over the previous study reported here warrants publication in a general interest journal or would be better published in a more specialized journal. Ultimately the editors have to decide that.

As for specific concerns, there are only a few:

- Clonal analysis with true null alleles should be performed for a select group of genes (namely pros and mirr) to allay concerns regarding potential ambiguity and off targets of RNAi experiments. The use of additional RNAi lines targeting the same genes would also be useful to confirm the findings.

- the authors over-interpret the results of the EC-specific perturbations (Fig 4), arguing that the ECs over-expressing pros and mirr take on an EE 'identity'. It seems more likely that these transcription factors simply induce their expected target genes without changing over-all identity of ECs (these cells are for example clearly still polyploid). More characterization would have to be performed to confirm that these cells have switched identity.

- In some figures (Figure 5E for example) channels have to be separated further to provide a clear view of the reported changes (LacZ expression in 5E is not clearly visible in the GFP+ cells). This should be done where necessary throughout the manuscript.

We sincerely thank all reviewers for their time, thoughtful reading of our manuscript, and their constructive comments. In this revised manuscript, we have addressed all of the comments raised. Major changes include:

1. We have described the methods in details, and we affirm that all experiments were performed with flies at adult stages: we did not strictly-control the ages of the flies, but for the majority of knock-down or overexpression experiments, all crosses were performed at the permissive temperature and the enclosed 3-7 day old flies were then collected and shifted to the restrictive temperature for 7 days before analysis. For RNA-seq and DamID analysis, 5-7 day old flies were collected and shifted to the restrictive temperature for 7 and 2 days, respectively, before analysis.

2. We have generated transgenic reporter lines driven by the identified enhancers with Pros-binding motif deleted ($NPF^{pros-\Delta motif-lacZ}$ and $CCHa1^{pros-\Delta motif-lacZ}$), and found that the loss of this motif totally abolished their expression in EEs. These data further supports the notion that Pros directly regulates neuropeptide gene transcription via its binding motif on the identified enhancers.

3. We have generated additional reporter lines ($Ptx1^{pros-lacZ}$ and $Fer1^{pros-lacZ}$) and demonstrated their responsiveness to Pros. Along with the previously-obtained Pros-DamID hits, our results further reinforce the idea that Pros directly regulate the transcription of both 2^o and 3^o TFs.

4. We have provided a schematic figure (Figure 7) to illustrate our hierarchical model and the three part code principle underlying EE subtype identities. This should be greatly helpful for the readers, and for guiding future research on this topic.

Minor changes include language improvement, panel reorganizations for Figures, especially for Figures 5 and 6, and the addition of separate channel images in several figures.

Please find our point-by-point responses below.

Reviewer #1 (Remarks to the Author):

This paper from Guo, Xi, and colleagues extends the current understanding of enteroendocrine (EE) cell specification in the Drosophila gut by focusing on the actions and interactions of a set of transcription factors that determine how ~10

different EE identities are established as the progeny of EE progenitors differentiate. The genetic analysis used is similar to what was done for fly neural stem cells more than a decade ago, and the general result is not dissimilar: EE progeny are subject to hierarchical regulation by different combinations of transcription factors (TFs) that determine specificity at higher or lower levels. This is not unexpected, and indeed several publications already define the different EE subtypes (by hormone expression combinations) and report some details of their specification. However this current analysis adds many details, including data on new TFs, and illustrates the logic of specification in a much more comprehensive way. It's a good quality analysis and overall I liked it very much. It will provide a useful framework to guide future studies of EE specification in *Drosophila* as well as mice and humans. I have a number of relatively minor comments the authors should address as they revise the paper, most of which pertain to adding details such that the description of the analysis is more accurate and complete.

We thank the reviewer for these positive remarks.

1. In most experiments a temperature sensitive Gal80-ts gene is used to control the time of RNAi or gene target activation. Please add details of when the TS-Gal80 was used, and when not, and also provide the timing of temperature shift and subsequent incubation for each experiment. This info can be denoted in the figures and figure legends.

Done as suggested. We have made notes in the figures, legends and/or main text about when and how Gal80^{ts} mediated gene knocking down or overexpression was conducted for each experiment.

2. The text has many grammatical mistakes, some of which obscure meaning. These should be corrected.

We have gone through the entire manuscript and corrected all of the language mistakes that we could identify.

3. The Results section often mentions information from other publications without citing them. Please make sure all references to previously published data include citations.

We have added references in several places as suggested.

4. On page 8 line 4, and Fig 2D "pros target genes" are described. But it is not clear how this subset of genes was defined. Please give a precise description of out this pros target gene subset was defined.

The following parameters were used to define Pros binding targets: 1) annotated oDam peak (oDam-Pros vs oDam) with log₂FoldChange is >1; 2) the adjusted P value is less than 0.01 (2 oDam-Pros replicates vs 3 oDam control replicates).

As for the “pros target gene subset” displayed in Figure 2d, these genes were putative Pros-binding targets, and were also significantly downregulated or upregulated in pros-depleted EE cells. We added this definition in the revised manuscript (page8, line 11-13).

5. The reporter genes described first on page 8 (2nd paragraph) and in Fig 2H-M are very nice. However the authors should describe these more precisely and provide maps of them.

We have added more details about these reporters of Pros target sites in the main text. In addition, we have also noted the location and the length of putative binding site on the maps shown in Figure 2.

6. It could be very informative to make versions of these reporter genes with the putative pros binding sites mutated, and analyze their expression in various situations.

Thanks for this insightful suggestion. Using the Pros binding motif (T-A/T-A-G-A/C/G-C-G/A/T) described previously (Cook et al., 2003; Choksi et al., 2006), we deleted 2 Pros binding motifs found in NPF (Motif 1: TTAGCCG, -1589bp to -1595bp of the TSS; Motif 2: TAAGCTG, -1471bp to -1465bp of the TSS) and the only motif found in CCHa1 (TAAGGCA, +1734bp to +1740bp of the TSS) enhancers, and generated transgenic reporters driven by the mutant enhancers (referred to as NPF^{pros-Δmotifs}-lacZ and CCHa1^{pros-Δmotif}-lacZ, respectively). As is shown in Fig.2h and Fig.2l of the revised manuscript, the deletion of the Pros motif abolished the expression of these reporters in EEs. This result reinforces the idea that these neuropeptide genes are directly regulated by Pros.

7. Although it is known that different EE subtypes have distinct regional distributions in the fly midgut, **regional variations in EE type and specification mechanisms** are rarely mentioned in this paper. The authors should try to include this information. At least, they should **indicate which regions were assayed to reach the conclusions that they make**. EE specification mechanisms are probably different in different midgut regions, and the authors do not acknowledge this issue sufficiently.

Agreed and in the revised manuscript, the regional information was

denoted in all figures and in the text.

8. On page 15, the authors describe how the Fer1 TF affects EE specification and the expression of their CCHa1-pros-LacZ reporter. This is quite interesting, but the results would be even better if they were able to predict the Fer1 binding site in the CCHa1 target gene, and test their prediction by mutating the site(s) in their CCHA1 reporter.

This is a great point. But due to limited genomic data, the bioinformatics analysis was not very helpful in predicting the Fer1 binding motif. We therefore performed Fer1 Dam-ID analysis to identify its binding regions. As is shown in Fig. 6a , the Fer1 binding region on CCHa1 locus is nicely matched with the Pros binding region, indicating that Pros and Fer1 binding sites are in close proximity within the CCHa1 enhancer. This is in agreement with our previous observation that the expression of CCHa1^{pros}-LacZ reporter in EEs was also dependent on Fer1, in addition to Pros.

Moreover, we found that the overexpression of Fer1 in EE cells was able to ectopically induce the expression of the CCHa1^{pros}-LacZ in R3 EE cells (Fig.6b-c). These observations collectively suggest that the CCHa1^{pros} enhancer region harbors both Pros and Fer1 binding motifs, and Pros and Fer1 exerts their regulatory role on CCHa1 expression via this common enhancer region.

9. The paper very much needs a summary figure that clearly lays out the hierarchy of factors that control the different EE identities. Such a figure will help guide future research on this topic.

Agreed and we have added a summary figure (Fig. 7), which includes a hierarchy model as well as a three-part TF combination model. This figure illustrates the principles underlying the cellular diversities of EEs discovered in this work and should be informative in guiding future research on this topic.

Reviewer #2 (Remarks to the Author):

Guo et al present an interesting study on enteroendocrine cell specification and plasticity. They examine the role for Prospero in EE specification and identity maintenance and discover presumptive Pros downstream genes and binding sites to include both the two class specific TFs as well as some subtype specific TFs. Further, they explore the post differentiation plasticity of

the EEs by depleting or overexpressing class specific TFs.

While overall the data is compelling and worth publishing, additional experiments/controls need to be performed to justify the conclusions. This is especially true regarding Prospero's role in EE identity maintenance and expression of subtype specific genes. Moreover, the conclusion that the authors make that there is a top-down hierarchical TF cascade doesn't fit their data. While Pros seem likely to control the 2nd tier TF (with notch binary sister fate), the 2nd tier do not directly control 3rd tier TFs. In particular, subtypes reflect regional identity, suggesting a combinatorial mechanism (rather than hierarchical).

We thank this reviewer for their positive remarks and insightful criticism. We will respond to the specific points in details below, but in regard to the identity maintenance function Pros, we probably did not accurately describe our methods in the previously-submitted version of the manuscript, as all the experiments were done in adult Drosophila. In addition, normally ISCs are mostly nonproliferating or slow-proliferating in young intestine, and the intestinal epithelium is replaced in a very slow manner, about once every 2 weeks or more (Jiang et al, 2009; Biteau et al., 2011), and the EEP cells (DI+ &Pros+) in adult midgut are relatively rare. Therefore, the observed EE fate switch phenotype cannot be simply explained by the regeneration of new EEs from ISCs in replacement of pre-existing EEs. The additional evidence supported by cell lineage tracing experiments will be described below.

We agree that the 2nd TFs do not regulate 3rd TFs in a hierarchical manner, but we believe the "hierarchical model" is still informative as it not only refers to the regulatory-hierarchical relationships between Pros and 2nd plus 3rd TFs, but also refers to functional-hierarchical relationships among Pros, 2nd TFs and 3rd TFs: the function of 3rd TFs is dependent on Pros, 2nd TFs, but not vice versa. We have now provided a schematic to describe the hierarchical model (Fig. 7).

Major points:

- 1) Much of the methodology is not stated, making it difficult to understand and interpret the data.
 - a. For example, in Figure 1A-G) there is no explanation of how the experiments were performed.
 - i. How long is the temperature shifted before analysis?
 - ii. What stage is analyzed?
 - iii. How is Pros depleted? Presumably UAS-Pros-RNAi, but it is not stated in the results, legends or methods.

Thanks for pointing out this important issue. We have thoroughly checked the manuscript and added details in methodology wherever is necessary. For Figure 1a-g, we used ProsV1-Gal4^{ts}, UAS-GFP; UAS-pros-RNAi flies. We performed parental crosses at 18°C, and the 3-7 day old females were then transferred to 29°C for 7 days before dissection. Two independent RNAi lines for pros were used in this experiment, and they give rise to identical phenotypes.

2) The conclusions about Pros maintaining EE identity are not convincing.

a. prosV1Gal4^{ts} is a problematic reagent to deplete Pros only from mature cells. 1) Pros is expressed starting from an immature stage. 2) Pros depletion may affect the expression of the prosV1Gal4^{ts}.

i. Are there other reagents to specifically target the mature EE state?

Pros transcripts are detected at low levels in EEPs and high levels in mature EEs, similar to CG32547 and TK or AstC (Guo et al, 2019). Therefore, the currently available EE drivers, including prosV1-Gal4 (pan-EE), CG32547-Gal4 (pan-EE) as well as TK/ AstC-Gal4 (about half of EEs), should all drive gene expression in EEPs. But we believe this does not cause a major issue when these lines are used to assess the function of mature EEs, as EEPs, marked by DI⁺Pros⁺, are very rare cells in the midgut. Based on the sorted EEs from adult midgut that are subjected to scRNA-seq analysis (Guo et al., 2019), the ratio of EEPs to mature EEs is less than 1% (43 EEPs vs. 4618 mature EEs). Therefore, the vast majority of the cells marked by prosV1-Gal4 should be differentiated/ mature EEs.

As shown in Fig.1g, and Fig. S1, the number of GFP⁺ cells driven by the pros-V1Gal4^{ts} driver were not significantly changed during the process of pros-depletion (from 1d to 7d) showing that the prosV1Gal4^{ts} driver remains active despite the cells have lost Pros protein, indicating that although the Pros-depleted EEs have lost many features of EEs, such as the loss of neuropeptide expression, these cells still retain certain memory, as the transcriptional activity of pros is still remained. Consistent with this observation, the expression of Pros protein and neuropeptides rapidly reappeared upon shifting back to the permissive temperature, (Fig.S1e-f).

b. While figure 1A-G methods are not described, based on the RNA sequencing methods section, it seems likely that Pros depletion was performed via a 7-day temperature shift at 29 degrees (expressing pros-RNAi). If this is the case, it is difficult to believe that the same “mature” EE cells can be visualized 7 days later. It is very possible the GFP⁺ cells are actually born post

Pros depletion-making them immature EE cells. The ISC division rate and EE to ISC ratio (Marianes and Spradling) suggest that many (if not most) GFP+ cells after 7 days at 29 could have been born after initiation of Pros depletion. I. The time course in Figure S1 doesn't really address these concerns as it is presented. Rather than "a gradual loss of Pros expression... [ac]companied by a gradual loss of hormone expression," a gradual loss of Pros+ and Astc+ cells is observed. This could easily be explained by cellular turnover (cell death/increase of new EEs).

We apologize for the confusion caused. Although we did not strictly-control the age of flies subjected to the experiments, all experiments were performed at adult stages: the parental crosses were performed at 18°C and for most experiments, we collected adult flies in the range of 3 to 7 days old, and these adult flies were then shifted to 29°C for 7 days before analysis.

As mentioned earlier, EEP represents a rare population of cells in the adult midgut. This is probably due to relative quiescence of ISC in young flies, as well as the very transient nature of EEPs, which divide and differentiate immediately once appear (Chen et. al, 2018). We think the ISC division activity was over-estimated by Marianes and Spradling: it is well-known that the heat shock treatment at 37°C can effectively induce ISC division.

Anyway, to further evaluate the relative contribution of the newly-generated EEs from ISCs and pre-existing EEs to the observed total EEs following the 14-day process of temperature shift procedures, we performed cell lineage tracing analysis using the control ProsV1^{ts}-Gal4 flies. After labeling EEs and tracing them for 7 days at 29°C, we then shifted them back to 18°C for 7 days before dissection and analysis. We found that the majority of Pros⁺ cells were labeled with the lineage marker (more than 90%, see the images and graph below), showing that potential contribution of the newly-generated EEs from ISCs is very small, if any.

c. In the time course (FigS1) do the number of GFP+ cells change 1-3d after temperature shift (when a significant decrease in AstC+ cell numbers is seen)?

We calculated the density of GFP+ cells following 3 days of Pros depletion. There was no significant decrease of GFP+ cell number upon pros depletion. However, the percentage of AstC+ cells was dramatically decreased (Fig. S1 i-j). This further supports the idea that *pros*-depletion does not cause cell death of EEs, but instead cause the loss of many EE signature gene expression (such as AstC).

d. Apoptotic markers should be examined.

We stained the apoptotic marker Dcp-1, found that there was no significant increase of Dcp-1+ EE cells following pros depletion (Fig.S1 g-h, which reinforces the idea that depletion of pros in EEs does not cause cell death.

e. If 1d post temp shift is sufficient for a significant decrease in AstC expression, can peptide hormone expression also recover in a similarly short time-period?

We performed the experiment as suggested. After inducing Pros depletion for 7 days at 29°C, we then shifted them back to 18°C for 1 day and stained Pros and AstC. We found that Pros expression was recovered in more than 60% of GFP week/negative cells (see graphs below), and remarkably, the expression of AstC was also reappeared (Fig.S1f, j).

3) The same issues regarding the identity of the cells (as mature EEs lacking Pros) being analyzed also apply to the FACS sorted cells. Perhaps even more so, as these are 5-7day old flies that have been shifted to 29 for 7 days, suggesting that Pros has been depleted since prior to eclosion and thus

completion of adult gut development.

a. How does the interpretation of the data change if the cells analyzed are immature EEs, not yet expressing Pros?

We apologize again for the confusion caused. 5-7day old flies were collected and subsequently been transferred to 29°C for 7 days. Therefore the experiments were performed at the adult stage.

4) The Dam-ID (Pros-Dam) is directly driven by prosV1Gal4ts, making it highly expressed compared to traditional Dam-ID or TaDA which require very low expression levels to prevent oversaturation and potential binding artifacts.

a. Additional evidence needs to be provided that the EE TFs are Pros target genes (particularly those with not as significantly enriched binding peaks, like Ptx1).

The TaDA system clearly has its advantages. We happened to use the oDam system because this system was previously used in our lab and it worked very well in our hands. One advantage of the oDam system is that the function of the Dam-TF fusion protein can be tested before conducting subsequent experiment, as sometimes the tag can affect the function of the protein. Certainly overexpression has a risk of oversaturation that generates non-specific bindings. We used several approaches to minimize the impact caused by non-specific bindings: 1) we drive oDam-TF (such as oDam-Pros) expression for a short time period (only 48h) to minimize time-dependent protein accumulation; 2) the replicates of both control and experimental groups were performed, and the log 2FC between Dam-pros versus Dam control was stringently set (such as >1) and the positive peaks were selected only for those whose adjusted p value is 0.01 or less; 3) We further filtered the putative Pros target genes by combined analysis with the comparative RNA-seq data, and genes with both significant DamID peaks and significantly altered expression level following pros depletion were selected as Pros target gene candidates. 4) Using reporters and antibodies, we further validated regulatory role of Pros on several putative Pros target genes.

We took the suggestion and generated a lacZ reporter line of Ptx1 using the putative binding region (+2432bp to +3619bp of the TSS) of Pros on Ptx1 loci. As is shown in Fig.2, we observed Ptx1^{Pros}-lacZ signals in EEs, and its expression was abolished following pros depletion (Fig.2 p-r).

b. How is Pros knocked down in Figure 2F,I&L? could loss of LacZ be due to change in cell fate?

Pros knocking down in Figure 2f,i&l were carried out using ProsV1-Gal4^{ts}. As also replied to reviewer 1, to further address the direct regulatory role of Pros on its target genes, we generated mutant versions of reporters by depleting Pros binding motifs within these targeting regions, and expression of these reporters were abolished (Fig.2 h, l).

c. Does Pros overexpression cause increase in LacZ levels?

We found that if pros was overexpressed in EEs, there was no discernible increase of NPF-lacZ expression in lacZ⁺ EEs (see images below). This suggests that normally in mature EEs, the expression of Pros has already reached a saturate level.

5) Introduction is lacking—in particular:

- a. the nature of EE diversity (explain the neuropeptide hormones better)
- b. what is already known about the TF code of the different types of EE cells (work from their lab).
- i. How are they defining tertiary TFs?
- c. The role of Notch in class 1 v class 2-what is already known.

We have added a few more points in the introduction as suggested. For the smoothness in logic flow, we decided to define tertiary TFs in the results session (page 10, line 1-5) where the issue was about to brought up.

6) FIGURE 3

- a. The “cell lineage” experiments are not actually lineage tracing because they use dynamic GAL4 drivers.

We respectfully disagree with the reviewer as based on single cell RNA-seq data as well as antibody staining patterns, both Tk and AstC are continuously and specifically expressed in all Tk⁺ or AstC⁺ EEs, respectively, and as such, the Tk- and AstC-Gal4 drivers should be considered as reliable class II and class I EE drivers, respectively.

This conclusion is also well-supported by the results we obtained from cell lineage tracing experiments. We have labeled and traced Tk-Gal4 or AstC-Gal4 cells for 7 days, and we found that all of the lineage marker positive cells (lacZ+) were also co-marked with UAS-GFP/RFP as well as the corresponding Tk/AstC protein based on antibody staining (Fig.3 j, m). Therefore, Tk+ EEs remain as Tk+ EEs during this tracing period, similarly for AstC+ EEs. However, as we have described in this study, altering mirr expression causes dramatic affects in this cell identity fidelity.

b. Figure 3A-F and 3J-M, single-color images should be shown.

Done as suggested.

c. Again, details are not given about how long genes are depleted or overexpressed is it 7 days?

Yes. For most experiments, genes were depleted or overexpressed for 7 days and details have now been added in the main text and/or denoted in the Figures.

d. there seems to be cell loss (see below)

i. In Mirr-IR see lots of EE doublets expressing AstC, but with UAS-mirr fewer doublets are obvious and those clear doublets appear to have one cell expressing TK and the other not....Are there the same # of GFP+ cells with UAS-mirr as control or mirr-IR? They look like they may be fewer in number (zoomed out a little too, so perhaps UAS-mirr causes death?).

As is shown in the updated Fig.3e, by adding the single color channel of Tk staining, it is easier to see that these EE doublets are all Tk+ (yellow arrowheads). Normally the EE population size in the midgut is variable among individuals. We therefore calculated the number of GFP+ EE cells in control, mirr-RNAi and UAS-mirr guts. As is shown in Fig.S4c, despite a slight increase in GFP+ cells in mirr-RNAi guts, the increase was not statistically significant.

We also stained the apoptotic marker Dcp-1 after overexpressing mirr for 7d. No significant increase of Dcp-1+ EE cells in mirr overexpressed guts compared with control, indicating that UAS-mirr did not cause EE death (Fig. S6b-c).

ii. 3K (UAS-mirr) also appears to be missing doublets compared to 3J are there fewer EEs in 3K?

This is a very interesting point. We suspect that the overexpression of mirr might have a negative effect on EEP mitosis, but intriguingly, the total number of EEs was not significantly decreased. Again, the observed differences could be due to the variability of the ISC activity and EE population size among individual flies.

iii. There are clearly fewer EEs in 3M and no doublets.

By adding the single color channel of AstC staining (see the updated Fig. 3m), the doublets should now be easily visible.

7) FIGURE 4

a. Again more methods needed

Details are now added in the revised manuscript.

b. For 4I-O is a single Pros downstream gene enough to call it trans-differentiation? Do other not immediate Pros downstream EE specific genes need to be turned on to say “trans-differentiation?”

Agreed and we removed the term “transdifferentiation” entirely from the manuscript. Instead, we describe the effect of ectopic pros in enterocytes as “it is able to induce the expression of AstC but not Tk”.

8) FIGURE 5

a. First paragraph of “Pros cooperates with 2...” is very confusing.

We have changed it to “Pros and 2⁰ TFs cooperate with 3⁰ TFs to define subclass EE subtypes”.

b. The text says “As described previously, Pros directly regulated the transcription of both secondary and tertiary TFs,” yet the tertiary TFs have not been fully described or defined. There is not definitive evidence that Pros directly regulates tertiary TFs.

The DamID experiment helped to identify a number of 3⁰ TFs as potential Pros target genes, including Drm, Hbn, Poxn, Fer1, Sug and Dac (Fig.2d, Fig. S8a). Picking Fer1 as an example, we found that knocking down Pros leads to significant down regulation of its expression (Fig. 2d). To further evaluate a direct role for Pros on Fer1 expression, we generated a lacZ reporter using the predicted binding region, and assessed its expression. As is shown in Fig.S8 b-c, Fer1^{Pros}-lacZ expression was detected in EEs at R4, and its expression was abolished upon pros knockdown. These

data further supports the notion that Pros directly regulates Fer1 and likely other tertiary TFs as well.

Reviewer #3 (Remarks to the Author):

This is an interesting follow-up study to a study published in Cell Reports by the same authors in which the authors had used scRNAseq to identify a transcription factor code that presumably controls enteroendocrine cell diversity in the Drosophila midgut. Here the authors test this hypothesis by performing a series of genetic perturbations combined with RNAseq and DamID that reveal TF dependencies in subtype maintenance in the enteroendocrine lineage.

Some of the findings reported here have been reported in the previous study (such as that Mirr knockdown inhibits Tk expression while Ptx1 knockdown inhibits AstC expression), but other convincingly make the case that subtype identities are maintained by combinatorial gene regulation by Pros, Ptx1/Mirr, Esg/Fer. As such it is an interesting study reporting convincing results from well-performed experiments. A major editorial question is whether the advance over the previous study reported here warrants publication in a general interest journal or would be better published in a more specialized journal. Ultimately the editors have to decide that.

We thank this reviewer for their positive remarks on this work as well as their concerns. We agree that the current study is built upon our previously published work and some results are seemingly “old” findings. To better demonstrate the novelty of this work, as also suggested by other reviewers, we have added a schematic figure (Fig. 7) in which the central findings of this work, including Pros as a selector protein, the hierarchical model and the combination principles of the three-part TF code underlying EE diversity are illustrated. With this, we hope that the reviewer could be convinced that our findings represent a significant step forward, which should provide an important foundation for guiding future basic and applied research on EEs.

As for specific concerns, there are only a few:

- Clonal analysis with true null alleles should be performed for a select group of genes (namely pros and mirr) to allay concerns regarding potential ambiguity and off targets of RNAi experiments. The use of additional RNAi lines targeting the same genes would also be useful to confirm the findings.

This is truly a critical issue when interpreting results solely based on RNAi approaches. As for Pros, this gene has been well established for

requirement of EE fate specification from ISCs, and MARCM clones induced by its null mutant *pros*^{JF17} has been found to be loss of Tk⁺ EE cells (Wang et.al, 2015). Here, as is shown below, we also found a complete loss of Pros⁺ EE cells, as well as peptide hormone AstC and Tk producing cells in MARCM clones of *pros*^{JF17}.

We also used two independent *pros*-RNAi lines (JF02308, Chr.3; HMJ02107, Chr.2) to induce *pros* depletion in EE cells using ProsV1-Gal4, leading to similar phenotypes: near complete loss of Tk⁺ and AstC⁺ cells, while no significant effect on the number of ProsV1>GFP⁺ cells (Fig.1 b-e for JF02308, Fig.1 c-f for HMJ02107).

As for *mirr*, we used two independent RNAi lines (JF02196, Chr.3; SH05171.N, Chr.2) to deplete its expression, and test its role on Tk⁺ Class II and AstC⁺ Class I EE fate. As is shown in the revised manuscript, knocking down *mirr* using either RNAi lines induces a significant downregulation of Tk⁺ EE cells and an increase of AstC⁺ cells (Fig.3 c-d for JF02196, Fig.S4a-b for SH05171.N). In addition, the exact opposite results were found when *mirr* was ectopically expressed.

- the authors over-interpret the results of the EC-specific perturbations (Fig 4),

arguing that the ECs over-expressing pros and mirr take on an EE 'identity'. It seems more likely that these transcription factors simply induce their expected target genes without changing over-all identity of ECs (these cells are for example clearly still polyploid). More characterization would have to be performed to confirm that these cells have switched identity.

Agreed and as also replied to the reviewer 2, we have removed the term transdifferentiation entirely from the manuscript.

- In some figures (Figure 5E for example) channels have to be separated further to provide a clear view of the reported changes (LacZ expression in 5E is not clearly visible in the GFP+ cells). This should be done where necessary throughout the manuscript.

Done as suggested.

REVIEWERS' COMMENTS

Reviewer #2 (Remarks to the Author):

Well done

Reviewer #3 (Remarks to the Author):

The authors have responded appropriately to my concerns, adjusting the interpretation of the results and adding additional genetic controls. The study can be recommended for publication as is.